# EEG state-trajectory instability and speed reveal global rules of intrinsic spatiotemporal neural dynamics

**Melisa Menceloglu**[1], **Marcia Grabowecky**[1,2], **Satoru Suzuki**[1,2]*

**1** Department of Psychology, Northwestern University, Evanston, Illinois, United States of America,
**2** Interdepartmental Neuroscience, Northwestern University, Evanston, Illinois, United States of America

\* satoru@northwestern.edu

**Data Availability Statement:** The link to the data repository is: https://doi.org/10.21985/n2-h6z9-3y27

**Funding:** National Institutes of Health T32 NS047987 to MM.

## Abstract

Spatiotemporal dynamics of EEG/MEG (electro-/magneto-encephalogram) have typically been investigated by applying time-frequency decomposition and examining amplitude-amplitude, phase-phase, or phase-amplitude associations between combinations of frequency bands and scalp sites, primarily to identify neural correlates of behaviors and traits. Instead, we directly extracted global EEG spatiotemporal dynamics as trajectories of $k$-dimensional state vectors ($k$ = the number of estimated current sources) to investigate potential global rules governing neural dynamics. We chose timescale-dependent measures of trajectory instability (approximately the 2nd temporal derivative) and speed (approximately the 1st temporal derivative) as state variables, that succinctly characterized trajectory forms. We compared trajectories across posterior, central, anterior, and lateral scalp regions as the current sources under those regions may serve distinct functions. We recorded EEG while participants rested with their eyes closed (likely engaged in spontaneous thoughts) to investigate intrinsic neural dynamics. Some potential global rules emerged. Time-averaged trajectory instability from all five regions tightly converged (with their variability minimized) at the level of generating nearly unconstrained but slightly conservative turns (~100° on average) on the timescale of ~25 ms, suggesting that spectral-amplitude profiles are globally adjusted to maintain this convergence. Further, within-frequency and cross-frequency phase relations appear to be independently coordinated to reduce average trajectory speed and increase the variability in trajectory speed and instability in a relatively timescale-invariant manner, and to make trajectories less oscillatory. Future research may investigate the functional relevance of these intrinsic global-dynamics rules by examining how they adjust to various sensory environments and task demands or remain invariant. The current results also provide macroscopic constraints for quantitative modeling of neural dynamics as the timescale dependencies of trajectory instability and speed are relatable to oscillatory dynamics.

**Competing interests:** The authors have declared that no competing interests exist.

## Introduction

EEG/MEG offer non-invasive methods to study large-scale neural dynamics with millisecond temporal precision (see [1] for a review). For the purpose of investigating neural dynamics, EEG/MEG from each scalp site is typically decomposed into a time series of frequency components (e.g., by using a Morlet wavelet convolution). Temporal associations among spectral amplitudes and/or phases are then examined within each site or between pairs of sites within the same frequency bands (cross-site only) or across different frequency bands (within- and cross-site). Temporal associations have also been examined between the phase of a lower-frequency band (typically $\theta$-$\alpha$) and the amplitude of a higher-frequency band (typically $\gamma$) either within or across sites. These analyses using time-frequency decomposition have elucidated how amplitude-amplitude, phase-phase, and/or phase-amplitude couplings within and across sites in different brain regions may contribute to various perceptual, attentional, and cognitive functions (e.g., [2–11]).

The current study was motivated by a complementary goal. Instead of trying to identify various types of oscillatory coupling as neural correlates of mental activities, we investigated general rules that may govern large-scale spatiotemporal dynamics. We focused on "intrinsic" dynamics reflected in EEG signals recorded while people rested with their eyes closed (while asked to allow spontaneous thoughts). Many EEG/MEG studies (also fMRI studies based on inter-region temporal correlations of BOLD signals on a much slower timescale) have examined what is called resting-state activity for diagnostic classification (e.g., [12–14]) as well as for identifying intrinsic spatial networks of phase-phase, amplitude-amplitude, and/or phase-amplitude couplings within or between specific frequency bands (e.g., [15–20]). While these studies have revealed much in terms of spatial distributions and graph-theoretic metrics of amplitude and phase relations, their primary goals were to target specific frequency bands to identify neural correlates of individual differences, dysfunctions, or tasks, or to distinguish spatial networks mediated by different frequency bands. The goal of the current study was complementary in that we aimed to elucidate general rules that may govern large-scale intrinsic neural dynamics by directly examining EEG spatiotemporal dynamics as multidimensional state vectors (without applying time-frequency decomposition).

We coded EEG from $k$ sites as a $k$-dimensional state vector $\overrightarrow{V_k}(t)$ (Fig 1A). Global EEG dynamics are entirely captured by the $k$-dimensional trajectories of a state vector. Large-scale neural processes are likely reflected in how these trajectories evolve. For example, a persistent trajectory movement in the same direction would indicate a continuation of the same process. The degree of directional instability of a trajectory (as a function of time $t$) for a specific timescale $\Delta t$ can be defined as the angular change $\theta$ in trajectory direction $\Delta \overrightarrow{V_k}$ (linearly interpolated over $\Delta t$ between $\overrightarrow{V_k}(t \pm \Delta t/2)$) across $\Delta t$ (i.e., between $\Delta \overrightarrow{V_k}(t \pm \Delta t/2)$). This measure is informative because the shape of a trajectory can be characterized based on the dependence of $\theta$ on $\Delta t$. For example, a smoothly curved trajectory is characterized by smaller $\theta$ values for a shorter $\Delta t$ and larger $\theta$ values for a longer $\Delta t$ (Fig 1B, left). A rough but overall straight trajectory is conversely characterized by larger $\theta$ values for a shorter $\Delta t$ and smaller $\theta$ values for a longer $\Delta t$ (Fig 1B, right). We thus define trajectory instability as,

$$Traj.Ins(t, \Delta t) = \theta(\Delta \overrightarrow{V_k}(t - \Delta t/2), \Delta \overrightarrow{V_k}(t + \Delta t/2)),\qquad \text{Eq 1}$$

where $t$ is time, $\Delta t$ is timescale, and $\theta$ is the angle between a pair of $(\Delta \overrightarrow{V_k})$s evaluated at $t-\Delta t/2$ and $t+\Delta t/2$, with $\theta = 0°$ indicating the continuation of a trajectory in the same direction, $\theta < 90°$ indicating the trajectory largely moving forward, $\theta = 90°$ indicating an orthogonal turn, and $\theta > 90°$ indicating the trajectory largely reversing (in $k$-dimensional space).

**A** $\overrightarrow{V_k}(t)$ : $k$-dimensional ($k$-site) state vector

**Trajectory instability** measured as the directional-change angle at time $t$ on timescale $\Delta t$,

$$Traj.Ins(t, \Delta t) = \theta\left(\Delta\overrightarrow{V_k}(t - \Delta t/2), \Delta\overrightarrow{V_k}(t + \Delta t/2)\right)$$

**Trajectory speed** at time $t$ on timescale $\Delta t$,

$$Traj.Spd(t, \Delta t) = \frac{\left|\Delta\overrightarrow{V_k}(t)\right|}{\Delta t}$$

**B** Red arrows indicate $\Delta\overrightarrow{V_k}(t \pm \Delta t/2)$

e.g. 1, a smoothly curved trajectory

e.g. 2, a rough but overall straight trajectory

**Stable** for shorter $\Delta t$ (= 2 intervals)

**Unstable** for shorter $\Delta t$ (= 2 intervals)

**Unstable** for longer $\Delta t$ (= 4 intervals)

**Stable** for longer $\Delta t$ (= 4 intervals)

**Fig 1. Illustrative examples of $k$-dimensional ($k$-site) state trajectories. A.** Definitions of the relevant variables, state vector, $\overrightarrow{V_k}(t)$, trajectory instability, *Traj.Ins*($t,\Delta t$), trajectory speed, *Traj.Spd*($t,\Delta t$), and timescale, $\Delta t$. **B.** The gray curves illustrate state trajectories with black dots indicating temporal sampling (512 per second). The red arrows illustrate $\Delta\overrightarrow{V_k}(t - \Delta t/2)$ and $\Delta\overrightarrow{V_k}(t + \Delta t/2)$, each linearly interpolated over $\Delta t$, used to compute trajectory instability at time $t$ on timescale $\Delta t$ in terms of the angle between them, $\theta$

(roughly curvature on timescale $\Delta t$). **Left.** A smoothly curved trajectory is characterized by smaller $\theta$ values for a shorter $\Delta t$ (e.g., $\Delta t = 2$ intervals) and larger $\theta$ values for a longer $\Delta t$ (e.g., $\Delta t = 4$ intervals). **Right.** A rough but overall straight trajectory is characterized by larger $\theta$ values for a shorter $\Delta t$ (e.g., $\Delta t = 2$ intervals) and smaller $\theta$ values for a long $\Delta t$ (e.g., $\Delta t = 4$ intervals).

In addition to the timescale dependence of trajectory instability, trajectory speed is informative,

$$Traj.Spd(t, \Delta t) = |\frac{\Delta \overrightarrow{V_k}(t)}{\Delta t}|, \qquad \text{Eq 2}$$

with a larger value indicating a greater rate at which the trajectory moves in the direction of $\Delta \overrightarrow{V_k}(t)$ on the timescale of $\Delta t$. Because trajectory instability and speed are essentially the second and first temporal derivatives of a multidimensional trajectory, analyzing them together allowed us to characterize the general properties of intrinsic EEG-state trajectories.

Our goal was to identify general rules governing intrinsic neural dynamics. Our strategy was to identify dynamical properties of EEG-state trajectories that are globally preserved. Specifically, we divided the "whole brain" (those aspects reflected in the current sources estimated from scalp EEG) into several regions that likely perform distinct computations in order to identify dynamical properties that remained invariant across those functionally distinct regions. Further, region specific differences in dynamical properties may elucidate how global rules may change to accommodate different functional demands.

We thus divided the 64 sites into five regions, posterior, central, anterior, left-lateral and right-lateral regions (see Fig 2) because prior research suggests that these regions may subserve broadly distinct processes that may require different dynamics (e.g., [21–27]). Overall, prior research suggests that EEG activity in the posterior region primarily reflects visual processes (including attention and working memory), activity in the central region primarily reflects auditory, tactile, linguistic, mnemonic, and motor processes (also including attention and working memory), and activity in the anterior region primarily reflects cognitive, regulatory, attentional, and decisional processes, while activity in any of these regions may be lateralized. Further, our recent study has demonstrated that these regions engage in distinct patterns of cross-frequency amplitude-amplitude and phase-amplitude interactions [28]. We thus applied the trajectory-instability and trajectory-speed analyses to the resting-state EEG recorded from these regions to identify the dynamical properties of EEG-state trajectories that remained invariant across all regions and those that were regionally altered to accommodate specific functional demands.

## Materials and methods

### Participants

Forty-eight Northwestern University students (33 women, 14 men, and 1 non-binary; ages 18 to 29 years, $M = 20.83$, $SD = 2.65$) gave informed consent to participate for monetary compensation ($10 per hour). All were right-handed, had normal hearing and normal or corrected-to-normal vision, and had no history of concussion. They were tested individually in a dimly lit room. The study protocol was approved by the Northwestern University Institutional Review Board.

### EEG recording procedures and preprocessing

EEG signals were recorded from 64 scalp electrodes at a sampling rate of 512 Hz using a Bio-Semi ActiveTwo system (using the default filter settings; attenuation to –3 dB at 1/5 of the

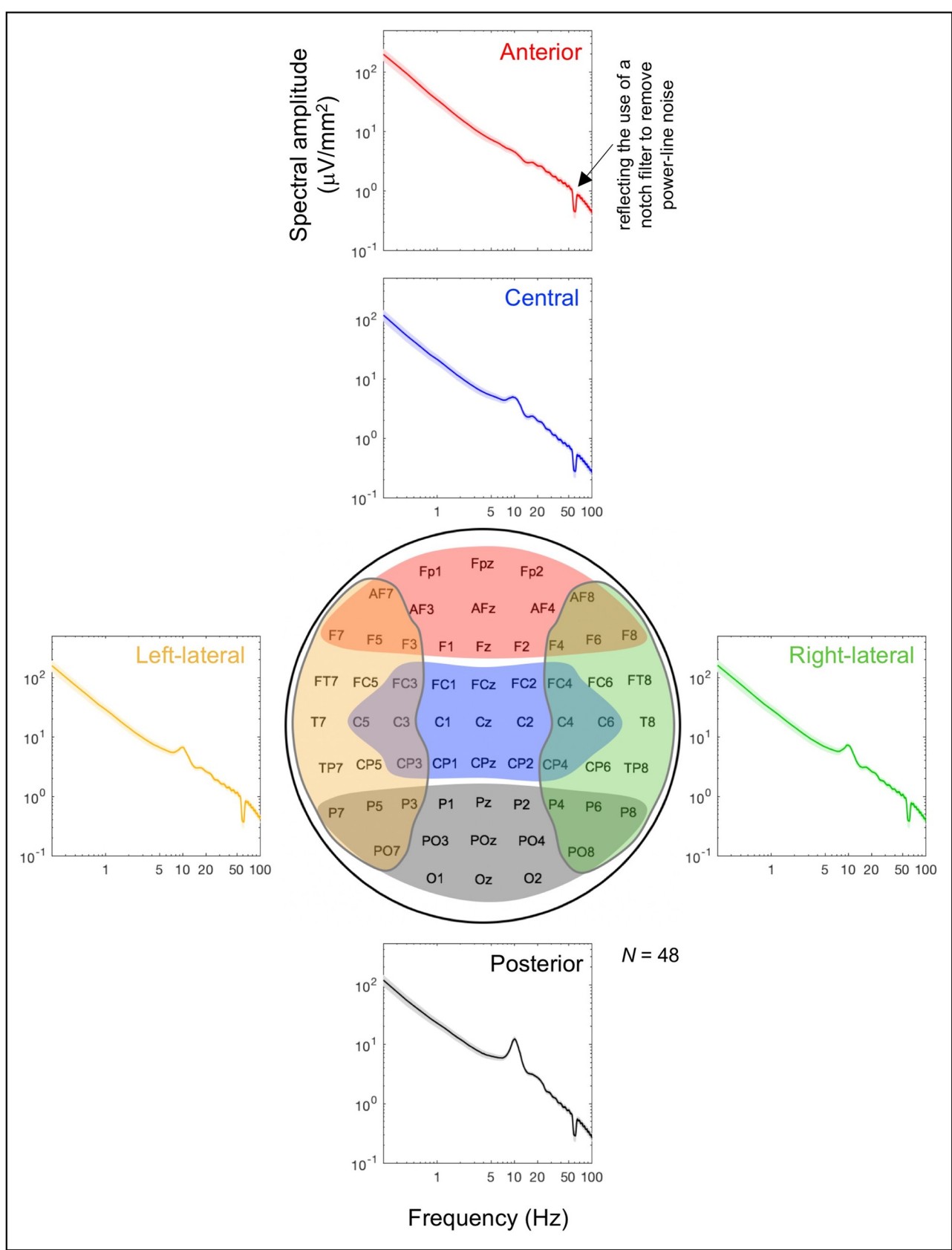

**Fig 2. Time-averaged spectral-amplitude profiles for the five regions for which EEG-state trajectory instability and speed were computed.** A time-averaged spectral-amplitude profile was computed for each region by applying a fast Fourier transform (FFT) to each 5-sec EEG segment and averaging the results across all segments per site per participant, then averaging the values across sites within each region and averaging across participants. The shaded areas represent ±1 standard error of the mean. The dips at 60 Hz reflect the use of a notch filter to remove power-line noise. The five regions, posterior (black), central (blue), anterior (red), left-lateral (orange), and right-lateral (green), are color coded and illustrated in the schematic electrode-site map in the middle.

sampling rate or 102.4 Hz; see www.biosemi.com) for about 5 minutes. The participants were instructed to rest with their eyes closed while minimizing movements and freely engage in whatever thoughts that spontaneously arose. We expect that the degree of mind wandering varied from participant to participant. This may have contributed some variability in our results given that resting fMRI-BOLD activity has been shown to vary when mind wandering is encouraged versus discouraged [29].

Two additional electrodes were placed on the left and right mastoid area. The EEG data were preprocessed with MATLAB using the EEGLAB and ERPLAB toolboxes [30–31]. The data were re-referenced offline to the average of the two mastoid electrodes, bandpass-filtered at 0.01–80 Hz, and notch-filtered at 60 Hz (to remove power-line noise that affected the EEG data from some of the participants). Based on our inspection of the EEG data, we did not identify any reasons to remove EEG signals over any time segments from any electrodes for any participants.

Because we compared EEG-state trajectory dynamics across regions, it was important to localize the underlying neural current sources from scalp-recorded EEG, especially to minimize any influences of volume conduction. To this end, we used a surface-Laplacian transform which has been shown to appropriately approximate the underlying current sources (especially the superficial sources near the skull) in a data-driven manner with minimal assumptions about the underlying sources (e.g., [32–34]). This transform also essentially removes effects of reference electrode choices and substantially reduces volume conduction effects largely confining them within adjacent electrodes for a 64-electrode montage like that used in the current study (e.g., [35]). We surface-Laplacian transformed the EEG data using Perrin and colleagues' method (e.g., [36–38]) with a typical set of parameter values [35]. Here we refer to surface-Laplacian transformed EEG simply as EEG.

## EEG data analysis

The continuous ~5 min EEG data (excluding the first 10 sec) were divided into consecutive 5-sec time segments and the first 58 segments from all participants were used for the analyses (some participants yielded 59 segments because their recording sessions were slightly longer). We chose to analyze the data in 5-sec segments for several reasons. First, as we focused on neural dynamics on the sub-second timescale, the segmenting allowed us to generate phase-scrambled control data (see below) for each 5-sec segment to control for any effects of gradual (slower than ~0.2 Hz) changes in spectral amplitudes. The choice of 5 sec was partly motivated by our recent finding that a half cycle of several seconds is a characteristic rate at which EEG spectral amplitudes fluctuate [28]. Second, we wanted to reliably characterize trajectory instability and speed up to the timescale of about a quarter of a second which corresponded to about a half-second cycle for the relevant underlying oscillatory activity (see below); thus, the use of 5-sec segments allowed us to include ~10 cycles across all timescales examined. We note in passing that the dynamical properties identified in the current study were relatively stable over a ~5 min period (see below), implying that the use of a longer analysis segment would have yielded comparable results.

While phase-scrambling can be performed using several different methods, we chose discrete cosine transform, DCT (e.g., [39]). In short, we transformed each 5-sec EEG waveform with type-2 DCT, randomly shuffled the signs of the coefficients, and then inverse-transformed it with type-3 DCT, which yielded a phase-scrambled version. DCT-based phase-scrambling is similar to DFT (discrete Fourier transform) -based phase-scrambling except that it is less susceptible to edge effects especially when phase-scrambling needs to be performed on relatively short waveforms.

We divided the whole head into five regions. The posterior region included, *Oz, O1, O2, POz, PO3, PO4, PO7, PO8, Pz, P1, P2, P3, P4, P5, P6, P7,* and *P8* (labeled with gray color in Fig 2); the central region included, *CPz, CP1, CP2, CP3, CP4, Cz, C1, C2, C3, C4, C5, C6, FCz, FC1, FC2, FC3,* and *FC4* (labeled with blue color in Fig 2); the anterior region included, *Fz, F1, F2, F3, F4, F5, F6, F7, F8, AFz, AF3, AF4, AF7, AF8, FpZ, Fp1,* and *Fp2* (labeled with red color in Fig 2); the left-lateral region included, *T7, FT7, TP7, F7, P7, C5, FC5, CP5, F5, P5, AF7, PO7, C3, FC3, CP3, F3,* and *P3* (labeled with orange color in Fig 2); the right-lateral region included, *T8, FT8, TP8, F8, P8, C6, FC6, CP6, F6, P6, AF8, PO8, C4, FC4, CP4, F4,* and *P4* (labeled with green color in Fig 2). Note that each region contained 17 sites. For measures of state-trajectory instability and speed to be commensurate across the five regions, the number of dimensions (i.e., the number of sites) needed to be matched across the regions.

The spectral amplitudes from the five regions showed typical profiles. Fast Fourier transform (FFT) was applied to each 5-sec EEG segment and the results were averaged across all segments per site per participant; the values were then averaged across sites within each region and averaged across participants. The resultant regional time-averaged spectral-amplitude profiles are shown in Fig 2. All regional profiles included the typical $1/f^b$ component that partly reflects an Ornstein-Uhlenbeck process which is a component of the standard integrate-and-fire model of a neuron, accounting for the stochastic membrane-potential dynamics in the absence of input currents (e.g., [40–41]; see [42] for a review of the various contributing factors). As expected from EEG recordings with the eyes closed, the posterior spectral-amplitude profiles included a pronounced $\alpha$ (~10 Hz) peak (the black curve in Fig 2) that was reduced in the central and left/right lateral profiles (the blue, orange, and green curves) and virtually absent in the anterior profiles (the red curve).

We represented the global spatiotemporal dynamics of each region as a 17-dimensional ($k = 17$) state vector $\overrightarrow{V_k}(t)$ with each dimension corresponding to EEG from a site within the region sampled at 512 Hz. State-trajectory instability and speed were evaluated as a function of time $t$ and timescale $\Delta t$ by first computing the state-vector differential $\Delta \overrightarrow{V_k}(t)$ (linearly interpolated over $t+\Delta t/2$) as a function of $t$ for a range of $\Delta t$ values. The smallest $\Delta t$ was two time-points (~3.9 ms at the sampling rate of 512 Hz), with subsequent $\Delta t$ values increased by integer multiples of ~3.9 ms, with the largest $\Delta t$ being 60 times as long as the smallest $\Delta t$, or ~234 ms. State-trajectory instability at time $t$ for a timescale $\Delta t$ was then evaluated as the angle $\theta$ between $\Delta \overrightarrow{V_k}(t - \Delta t/2)$ and $\Delta \overrightarrow{V_k}(t + \Delta t/2)$, which was computed as,

$$Traj.Ins(t, \Delta t) = acos\left[\frac{\Delta \overrightarrow{V_k}(t - \Delta t/2) \cdot \Delta \overrightarrow{V_k}(t + \Delta t/2)}{|\Delta \overrightarrow{V_k}(t - \Delta t/2)||\Delta \overrightarrow{V_k}(t + \Delta t/2)|}\right] \qquad \text{Eq 3}$$

State-trajectory speed at time $t$ for a timescale $\Delta t$ was computed as,

$$Traj.Spd(t, \Delta t) = \left|\frac{\Delta \overrightarrow{V_k}(t)}{\Delta t}\right|. \qquad \text{Eq 4}$$

Both trajectory instability and speed may depend on phase-independent spectral-amplitude profiles (e.g., reduced high-frequency amplitudes would generate smoother trajectories regardless of phase relations) as well as on phase relations. Comparisons with phase-scrambled controls (see above) allowed us to isolate the phase-dependent aspects of trajectory instability and speed indicative of active phase coordination. In particular, by using two types of phase scrambling (see Results and Discussion), we distinguished the contributions of cross-site within-frequency and cross-frequency phase relations to EEG-state trajectories. We examined the temporal average and standard deviation of trajectory instability and speed, as well as the temporal correlation between trajectory instability and speed, computed for both real data and the corresponding phase-scrambled controls as a function of $\Delta t$. These statistics were first computed for each 5-sec segment and then averaged across all segments per region per participant. We demonstrate statistical reliability with small error regions (indicating ±1 standard error of the mean with participants as the random effect) relative to mean differences, virtually identical patterns obtained from the odd and even numbered participants, and clearly separated distributions of values.

## Results and discussion

### Effects of spectral-amplitude profiles on trajectory instability and speed

The timescale dependence of EEG state-trajectory instability was characteristically different across the posterior, central, anterior, left-lateral, and right-lateral regions (Fig 3A). On the shorter timescales ($\Delta t < {\sim}25$ ms), the trajectories were most stable in the posterior region, intermediate in the central and left/right-lateral regions, and least stable in the anterior region, whereas the pattern flipped on the intermediate timescales ($\Delta t = 25{-}80$ ms). Notably, the levels of trajectory instability in all five regions tightly converged on the timescale of ~25 ms at an instability level of ~100˚. The tightness of convergence is indicated (1) by the narrow cross-participant distribution of the values of critical timescale $\Delta t$ (mostly within 20–30 ms; Fig 3B) that minimized the inter-region standard deviation in trajectory instability, (2) by the narrow distribution of the trajectory-instability levels (mostly within 90˚-110˚; Fig 3C) at the critical timescale, and (3) by the high precision of convergence, with the inter-region standard deviations of trajectory instability dipping to only 1˚-2˚ at the critical timescale for most participants (Fig 3D). The virtually identical patterns of timescale dependence of trajectory instability for the odd and even numbered participants (Fig 3E) demonstrate statistical reliability. We note that the tight global (inter-region) convergence of state-trajectory instability on the timescale of ~25 ms is primarily a characteristic of baseline spectral-amplitude profiles independent of phase relations (see below).

State-trajectory speed overall diminished with increasing timescale $\Delta t$ in all five regions (Fig 3F). In addition, the posterior region had the overall fastest trajectories, followed by the left/right lateral regions and the anterior region with the central region having the slowest trajectories. Further, the trajectory speeds in the posterior, left/right lateral, and anterior regions converged on the short ($\Delta t < {\sim}10$ ms) and long ($\Delta t > {\sim}130$ ms) timescales, while the trajectory speeds in the central region remained low across all examined timescales. These patterns are reliable as those obtained from the odd and even numbered participants were virtually identical (Fig 3G). We note that these patterns of state-trajectory speed are also primarily a characteristic of baseline spectral-amplitude profiles independent of phase relations (see below).

### Effects of phase relations on trajectory instability and speed

Comparing the real trajectory instability and speed with those computed using the corresponding phase-scrambled controls revealed trajectory characteristics that depended on

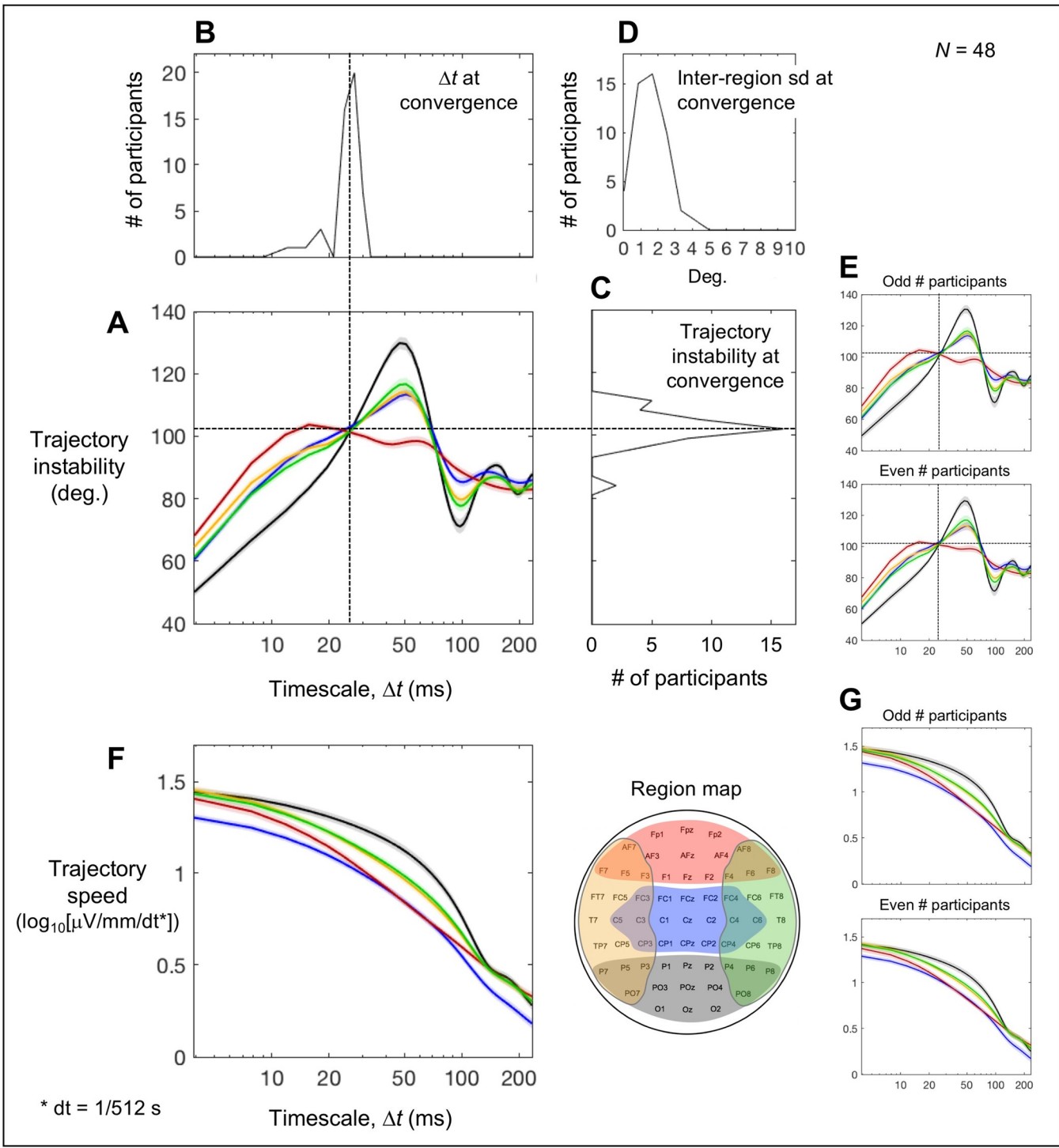

**Fig 3. Trajectory instability and speed as a function of timescale Δ*t*.** A-D. Trajectory instability. **A.** Time-averaged trajectory instability for the posterior (black), central (blue), anterior (red), left-lateral (orange), and right-lateral (green) regions as a function of Δ*t*. Notably, the five curves tightly converge at Δ*t* ~ 25 ms where the level of trajectory instability is ~100° (vertical and horizontal dashed lines). To illustrate the consistency of this convergence across participants, the cross-participant distributions (histograms) of the Δ*t* values and trajectory-instability levels at the convergence are shown in **B** and **C**, respectively. To illustrate the precision of convergence, the cross-participant distribution of the inter-region standard deviation (sd) of trajectory instability at the convergence is shown in **D** (mostly 1°-2° and all less than 5°). **E.** The time-averaged trajectory instability pattern in A shown separately for the odd and even numbered participants; the virtually identical patterns demonstrate statistical reliability. **F.** Time-averaged trajectory speed for the posterior (black), central (blue), anterior (red), left-lateral (orange), and right-lateral (green) regions as a function of Δ*t*. **G.** The time-averaged trajectory speed in F shown

separately for the odd and even numbered participants; the virtually identical patterns demonstrate statistical reliability. The shaded areas represent ±1 standard error of the mean.

spectral-phase relations over and above baseline spectral amplitudes (our phase-scrambled controls closely tracked slow variations in spectral amplitudes as we used 5-sec segments as the unit of analyses; see Material and Methods).

The phase-dependent variability in time-averaged trajectory instability was small for all regions, limited to less than ±2° (Fig 4A), indicating that the characteristic regional profiles of time-averaged trajectory instability including the tight convergence at $\Delta t \sim 25$ ms (Fig 3A) primarily reflect baseline spectral-amplitude profiles.

Nevertheless, phase relations produced some consistent effects on trajectory instability. The posterior region yielded increased instability (relative to the phase-scrambled control) on the short < 15 ms and longer ~100–140 ms timescales (the black curve in Fig 4A) with the latter peak accompanied by the left and right lateral regions (the orange and green curves). Further, all regions except the anterior region (red) yielded reduced instability on the mid-range ~30–80 ms timescales, all regions except the posterior region (black) yielded reduced instability on the short < 8 ms timescales, and all regions yielded reduced instability on the long > ~180 ms

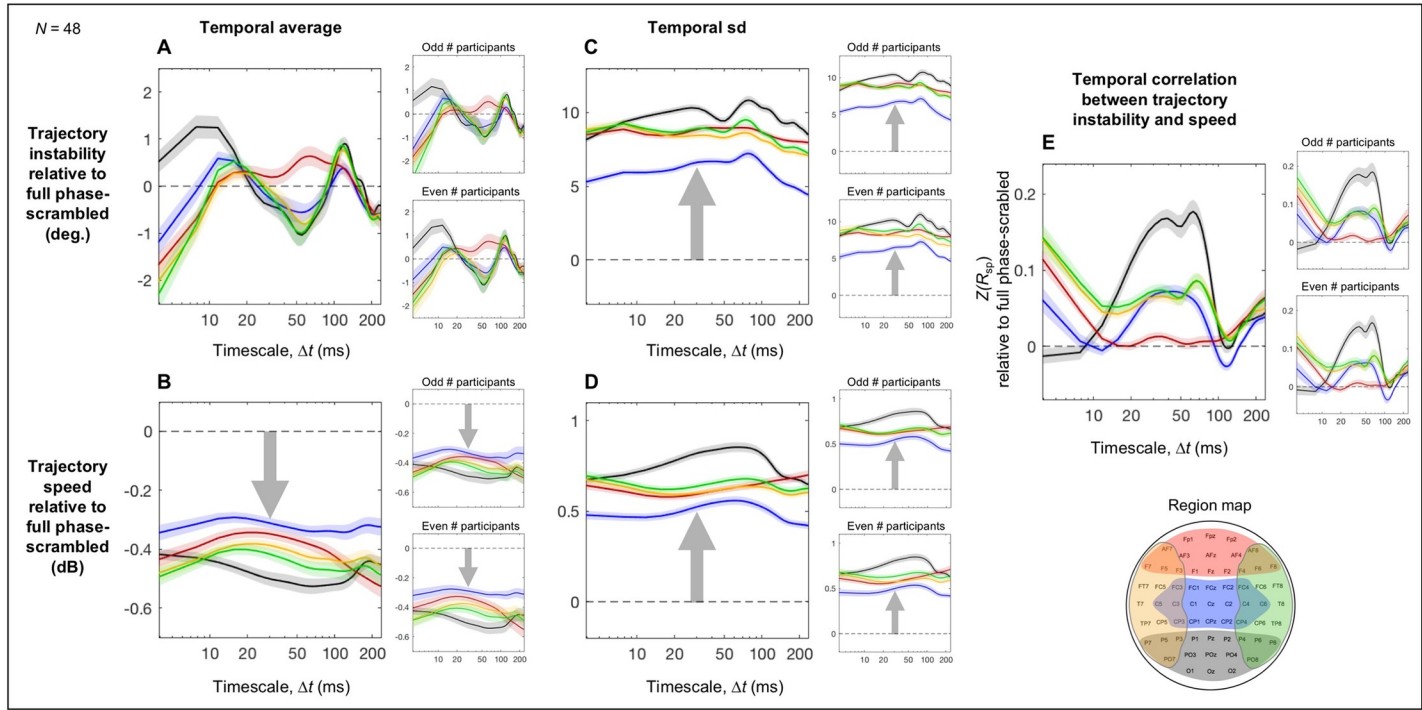

**Fig 4. The phase-relation dependent aspects of trajectory instability and speed and their temporal correlation as a function of timescale Δt relative to the corresponding full phase-scrambled controls.** For each plot, the reliability of the data patterns is demonstrated by the accompanying plots on the right showing virtually identical patterns obtained from the odd and even numbered participants. **A.** Time-averaged trajectory instability as a function of Δt (real data minus the corresponding phase-scrambled controls). **B.** Time-averaged trajectory speed as a function of Δt (real data vs. the corresponding phase-scrambled controls in dB). **C.** The temporal variability (measured as standard deviation) in trajectory instability as a function of Δt (real data minus the corresponding phase-scrambled controls). **D.** The temporal variability (measured as standard deviation) in trajectory speed as a function of Δt (real data vs. the corresponding phase-scrambled controls in dB). **E.** The temporal correlation between trajectory instability and speed as a function of Δt (real data minus the corresponding phase-scrambled controls in z-transformed Spearman r). In all panels, the data for the posterior, central, anterior, left-lateral, and right-lateral regions are shown in black, blue, red, orange, and green curves, respectively (see the region map). The shaded areas represent ±1 standard error of the mean.

timescales. These patterns were obtained from both the odd and even numbered participants (the right panels in Fig 4A).

While phase relations had overall small effects on time-averaged trajectory instability (Fig 4A), they substantially reduced time-averaged trajectory speed in all regions. Further, the levels of speed reductions in dB were largely invariant across all examined timescales in all regions (Fig 4B). This suggests that similar mechanisms of phase coordination divisively reduce trajectory speed in all regions. This also indicates that the characteristic regional profiles of time-averaged trajectory speed (Fig 3F) primarily reflect baseline spectral-amplitude profiles.

Active control of phase relations, which includes synchronization and desynchronization, likely influences temporal variability in trajectory instability and speed (see General Discussion for details). Accordingly, the temporal variability (measured as temporal standard deviation) in both trajectory instability (Fig 4C) and speed (Fig 4D) were substantially increased in all regions (relative to the phase-scrambled controls). Further, the substantial increases in variability (in degrees for the instability variability and in dB for the speed variability) were largely invariant across the examined timescales in all regions. Nevertheless, the variability in both trajectory instability and speed was most elevated in the posterior region, followed by the anterior and left/right lateral regions and least elevated in the central region (Fig 4C and 4D). These patterns were virtually identical for the odd and even numbered participants (the right panels in Fig 4C and 4D).

Given this evidence of coordination of trajectory instability and speed through modulation of phase relations, they may be jointly coordinated. Specifically, a positive temporal correlation between trajectory instability and speed would indicate faster trajectories being less stable (and slower trajectories being more stable), a negative correlation would indicate faster trajectories being more stable (and slower trajectories being less stable), whereas a lack of correlation would suggest independent control of trajectory instability and speed. We estimated the temporal correlations between trajectory instability and speed that reflected active spectral-phase coordination as the Fisher-Z transformed Spearman's $r$ values (robust against outliers) computed for the real data minus those computed for the phase-scrambled controls. Across all five regions trajectory instability and speed were overall either positively correlated or uncorrelated, implying that the dynamics in all regions are coordinated such that faster trajectories are less stable (and slower trajectories are more stable) (Fig 4E). In addition, all regions except the anterior region (red) yielded positive correlations on the ~20–80 ms timescales, all regions except the posterior region (black) yielded positive correlations on the fast < 8 ms timescales, and all regions yielded positive correlations on the slow > ~180 ms timescales. The lack of correlation for the anterior region (red) on the timescales of ~15–100 ms is notable, suggesting that trajectory instability and speed are independently controlled in the anterior region on these timescales. All these patterns were virtually identical for the odd and even numbered participants (the right panels in Fig 4E).

## Effects of within-frequency versus cross-frequency phase relations

Phase relations include both within-frequency phase relations—phase relations within the same frequency bands across sites—and cross-frequency phase relations—phase relations between different frequency bands within and across sites. The analyses presented in Fig 4 include the contributions of both types of phase relations because the phase-scrambling procedure randomized spectral phase independently for each frequency component at each site. We examined the unique contributions of cross-frequency phase relations using "cross-frequency phase-scrambled" controls, computed by independently randomizing phase for each frequency component but applying the same set of cross-frequency phase shifts to all sites. This

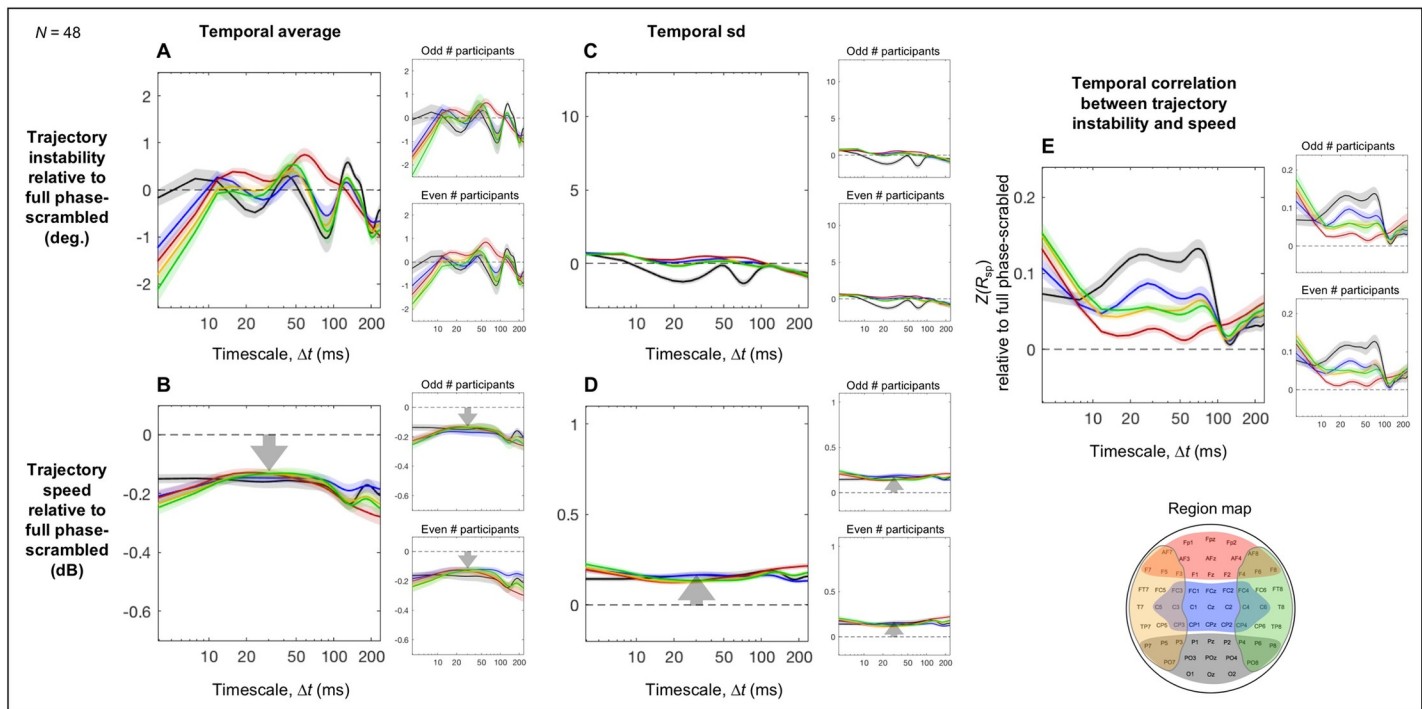

**Fig 5. The cross-frequency-phase-relation dependent aspects of trajectory instability and speed and their temporal correlation as a function of timescale Δ*t* relative to the corresponding cross-frequency phase-scrambled controls (that preserved cross-site within-frequency phase relations).** For each plot, the reliability of the data patterns is demonstrated by the accompanying plots on the right showing virtually identical patterns obtained from the odd and even numbered participants. **A.** Time-averaged trajectory instability as a function of Δ*t* (real data minus the corresponding phase-scrambled controls). **B.** Time-averaged trajectory speed as a function of Δ*t* (real data vs. the corresponding phase-scrambled controls in dB). **C.** The temporal variability (measured as standard deviation) in trajectory instability as a function of Δ*t* (real data minus the corresponding phase-scrambled controls). **D.** The temporal variability (measured as standard deviation) in trajectory speed as a function of Δ*t* (real data vs. the corresponding phase-scrambled controls in dB). **E.** The temporal correlation between trajectory instability and speed as a function of Δ*t* (real data minus the corresponding phase-scrambled controls in *z*-transformed Spearman r). In all panels, the data for the posterior, central, anterior, left-lateral, and right-lateral regions are shown in black, blue, red, orange, and green curves, respectively (see the region map). The shaded areas represent ±1 standard error of the mean.

procedure randomized cross-frequency phase relations within and across sites (e.g., X-Hz and Y-Hz components were phase-shifted by Δx and Δy degrees), but preserved within-frequency phase relations across sites (e.g., X-Hz components at all sites were phase-shifted by Δx degrees and Y-Hz components at all sites were phase-shifted by Δy degrees).

The effects of cross-frequency phase relations alone (measured relative to the cross-frequency phase-scrambled controls) were generally smaller than the combined effects of cross- and within-frequency phase relations (measured relative to the full phase-scrambled controls) (Figs 5 and 4). Nevertheless, some results are informative. Note that comparisons with the full phase-scrambled controls reflect combined influences of both cross-site within-frequency and cross-frequency phase relations, whereas comparisons with the cross-frequency phase-scrambled controls reflect influences of only cross-frequency phase relations. Thus, (1) a unique contribution of cross-site within-frequency phase relations may be inferred based on a reliable difference from the full phase-scrambled control with no reliable difference from the cross-frequency phase-scrambled control. (2) A unique contribution of cross-frequency phase relations may be inferred based on reliable but equivalent differences from both the full and cross-frequency phase-scrambled controls. (3) Joint contributions of cross-site within-frequency and cross-frequency phase relations may be inferred based on a large reliable difference from the full phase-scrambled control and a substantially smaller but still reliable difference from the

cross-frequency phase-scrambled control. We used this logic to interpret the results relative to the full and cross-frequency phase-scrambled controls.

For time-averaged trajectory instability, cross-site within-frequency phase relations uniquely contributed to increasing trajectory instability in the posterior region on the fast < 15 ms timescales as it was elevated relative to the full phase-scrambled control (the black curve in Fig 4A) but equivalent to the cross-frequency phase-scrambled control (the black curve in Fig 5A). Cross-frequency phase relations uniquely contributed to reducing trajectory instability in all regions except the posterior region on the fast < 8 ms timescales as the reductions were equivalent relative to both the full and cross-frequency phase-scrambled controls (the non-black curves in Figs 4A and 5A). Similarly, cross-frequency phase relations uniquely contributed to reducing trajectory instability in all regions on the slow > ~180 ms timescales as the reductions were equivalent relative to both the full and cross-frequency phase-scrambled controls (all curves in Figs 4A and 5A). All these patterns were virtually identical for the odd and even numbered participants (the right panels in Figs 4A and 5A).

For time-averaged trajectory speed, both cross-site within-frequency phase relations and cross-frequency phase relations contributed to its global (across all regions) and largely timescale-invariant reductions as the reductions were consistently larger relative to the full phase-scrambled controls than relative to the cross-frequency phase-scrambled controls (compare Figs 4B and 5B), but the smaller reductions relative to the cross-frequency phase-scrambled controls were still consistent (Fig 5B). These patterns were virtually identical for the odd and even numbered participants (the right panels in Figs 4B and 5B).

For the temporal variability in trajectory instability, cross-site within-frequency phase relations uniquely contributed to its global (across all regions) and largely timescale-invariant increases as the increases were substantial relative to the full phase-scrambled controls (Fig 4C) but virtually absent relative to the cross-frequency phase-scrambled controls (Fig 5C); if anything, cross-frequency phase relations contributed to reducing the temporal variability in trajectory instability in the posterior region on the ~15–40 ms and ~60–90 ms timescales (the black curve in Fig 5C). These patterns were virtually identical for the odd and even numbered participants (the right panels in Figs 4C and 5C).

For the temporal variability in trajectory speed, both cross-site within-frequency phase relations and cross-frequency phase relations contributed to its global (across all regions) and largely timescale-invariant increases as the increases were consistently larger relative to the full phase-scrambled controls than relative to the cross-frequency phase-scrambled controls (compare Figs 4D and 5D), but the smaller increases relative to the cross-frequency phase-scrambled controls were still consistent (Fig 5D). These patterns were virtually identical for the odd and even numbered participants (the right panels in Figs 4D and 5D).

For the temporal correlation between trajectory instability and speed, cross-frequency phase relations uniquely contributed to the patterns of positive correlations in the left/right-lateral and anterior regions across all examined timescales and in the central and posterior regions on the timescales slower than ~20 ms as those patterns were equivalent relative to both the full and cross-frequency phase-scrambled controls (Figs 4E and 5E); an exception may be that in the posterior region on the timescales of ~30–70 ms, the positive correlations were more elevated relative to the full phase-scrambled control (the black curve in Fig 4E) than relative to the cross-frequency phase-scrambled control (the black curve in Fig 5E) suggesting an additional contribution of cross-site within-frequency phase relations. In addition, in the posterior and central regions on the fast < ~15 ms timescales, cross-frequency phase relations increased but cross-site within-frequency phase relations reduced the positive correlations as they were more elevated relative to the cross-frequency phase-scrambled controls (the black and blue curves in Fig 5E) than relative to the full phase-scrambled controls (the black and

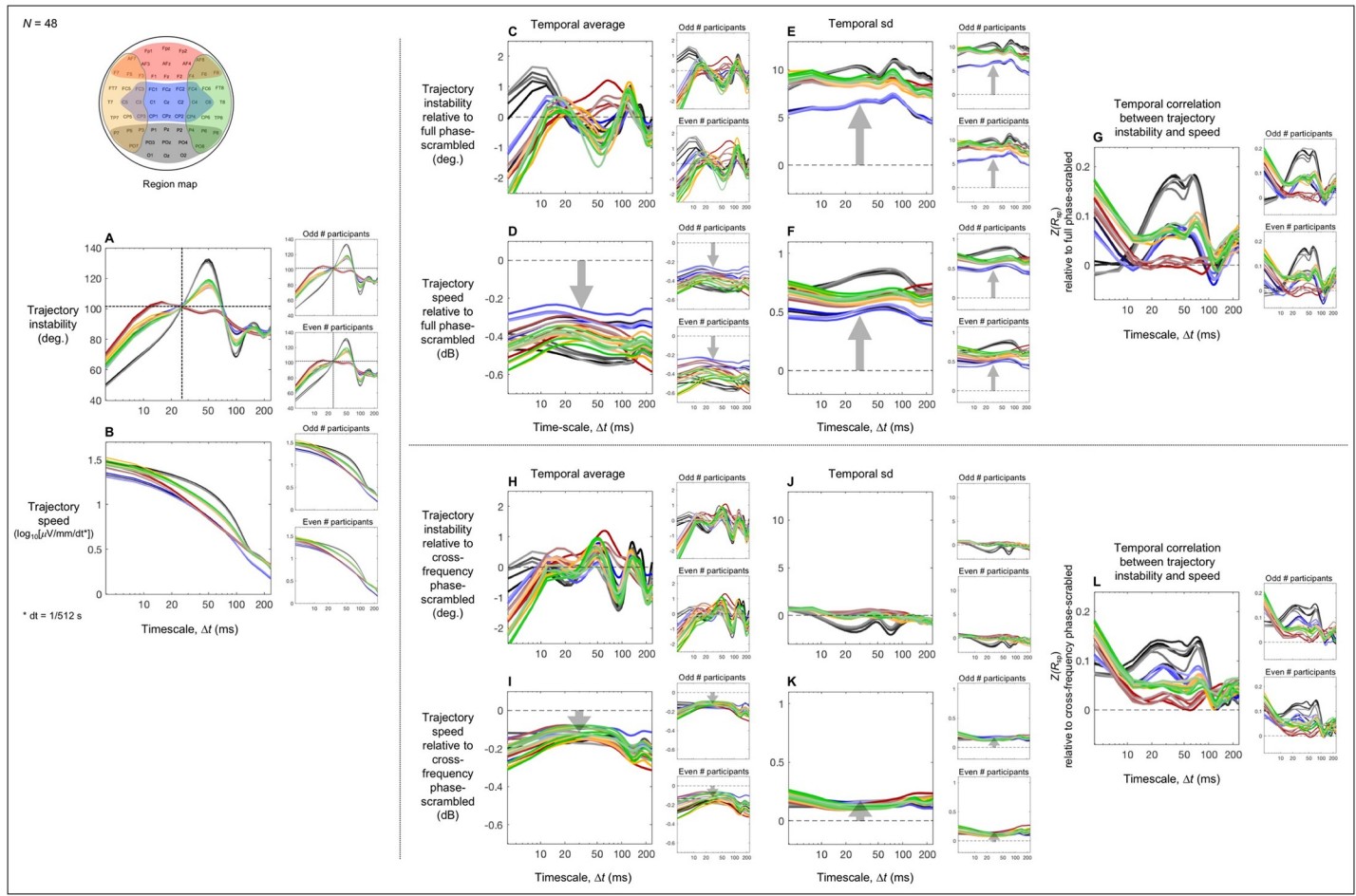

**Fig 6. The results shown in Figs 3–5 re-plotted separately for successive 50-sec intervals to show the temporal stability of those results over the ~5-min period.** Lighter shades indicate later intervals. **A-B.** Time-averaged trajectory instability and speed as a function of timescale Δ*t*, corresponding to Fig 3A–3F. **C-G.** The full-phase-relation dependent aspects of time-averaged trajectory instability and speed, their temporal variability, and their temporal correlations as a function of timescale Δ*t*, corresponding to Fig 4A–4E. **H-L.** The cross-frequency-phase-relation dependent aspects of time-averaged trajectory instability and speed, their temporal variability, and their temporal correlations as a function of timescale Δ*t*, corresponding to Fig 5A–5E. For each plot, the reliability of the data patterns is demonstrated by the accompanying plots on the right showing virtually identical patterns obtained from the odd and even numbered participants.

blue curves in Fig 4E). It is noteworthy that phase relations appear to be coordinated in such a way that trajectory instability and speed are uniquely independent in the anterior region on the timescales of ~15–120 ms as the correlations did not deviate from the full phase-scrambled control (the red curve in Fig 4E). All these patterns were virtually identical for the odd and even numbered participants (the right panels in Figs 4E and 5E).

Finally, we evaluated the temporal stability of the results presented in Figs 3–5 by computing the corresponding quantities separately in successive 50-sec intervals, dividing each ~5-min EEG recording period into 5 successive intervals. The data from the 5 intervals are shown as superimposed curves in Fig 6 with later intervals indicated with lighter colors. The time-averaged trajectory instability and speed as a function of Δ*t* (presented in Fig 3A and 3F), which primarily depend on phase-independent spectral-amplitude profiles (Fig 2), are replotted in Fig 6S and 6B. The full-phase-relation dependent aspects (relative to the full phase-scrambled controls) of the time-averaged trajectory instability and speed, their temporal variability, and their temporal correlations as a function of Δ*t* (presented in Fig 4A–4E) are

replotted in Fig 6C–6G. The cross-frequency-phase-relation dependent aspects (relative to the cross-frequency phase-scrambled controls) of these variables (presented in Fig 5A–5E) are replotted in Fig 6H–6L. It is clear from Fig 6 that all results are reasonably stable over a ~5-min period. Further, all the results are remarkably consistent for the odd and even numbered participants.

## General discussion

We investigated potential global rules governing large-scale intrinsic neural dynamics by characterizing the spatiotemporal dynamics of resting-state EEG as trajectories of $k$-dimensional state vectors where $k$ is the number of scalp sites (with each site corresponding to a current source estimated with the surface-Laplacian transform). We focused on the timescale dependence of state-trajectory instability and speed because these variables, approximately the 2nd and 1st temporal derivatives, are informative for characterizing general properties of state trajectories (e.g., Fig 1). We examined these trajectory properties in five broadly defined regions (posterior, central, frontal, left-lateral, and right-lateral regions) that likely perform distinct computations based on prior research including ours (e.g., [21–28]). Our strategy was to characterize the global rules governing intrinsic neural dynamics by identifying the trajectory properties that are preserved across all these regions. A secondary goal was to examine how those rules may be altered in separate regions to shed light on how global rules may change to support regional computational needs.

To interpret the current results in the form of timescale dependencies of EEG-state trajectory instability and speed, we consider how behaviors of these state variables relate to underlying neural spectral dynamics. Neurophysiological signals including scalp-recorded EEG have been known to exhibit characteristic spectral-amplitude profiles, generally including a $1/f^b$ background profile with embedded peaks at specific frequency bands such as $\theta$, $\alpha$, and $\beta$ bands (e.g., see [43] for a review; also see Fig 2 for examples from the current study). The $1/f^b$ background is thought to reflect random-walk type neuronal membrane fluctuations (in the absence of input currents) described by an Ornstein-Uhlenbeck process (Eq 5) included in the standard integrate-and-fire model (e.g., [40–41]),

$$\frac{dv(t)}{dt} = -\gamma v(t) + \sqrt{2D} \cdot gn(t), \qquad \text{Eq 5}$$

where $v(t)$ is the membrane potential, $\gamma$ is $1/\tau_m$, the reciprocal of the membrane time constant, $D$ is the noise intensity multiplied by $\gamma^2$, and $gn(t)$ is Gaussian noise. The membrane potential, $v(t)$, has a monotonically decreasing spectral-power profile, reducing to half power (or $1/\sqrt{2}$ amplitude) at the frequency, $f_{1/2} = \frac{\gamma}{2\pi}$, and approximating a $1/f^2$ power profile (or a $1/f$ amplitude profile) at higher frequencies (see [42] for a review of contributions of this and other mechanisms to the $1/f^b$ spectral-power profiles). Peaks at specific frequency bands are thought to reflect oscillatory neural activity, and their roles in coordinating processes that mediate perceptual, attentional, memory, and cognitive operations have been well studied (e.g., [2–11, 20, 44]). However, it has been less clear how the diverse spectral-amplitude profiles observed in different brain regions may be coordinated to maintain the global integrity of neural dynamics. The current results may contribute to an understanding of dynamical parameters relevant to such global coordination.

## Influences of spectral-amplitude profiles on EEG state-trajectory instability and speed

The time-averaged state-trajectory instability in the posterior, central, anterior, left-lateral, and right-lateral regions tightly converged on the timescale of ~25 ms at the instability level of ~100˚ (Fig 3A–3D). Because phase relations made minimal contributions to the time-averaged trajectory instability (less than ±2˚ across all examined timescales in all regions; Fig 4A), this convergence likely reflects a global coordination of baseline spectral-amplitude profiles. That is, the baseline spectral-amplitude profiles appear to be globally (across brain regions) coordinated in such a way that, while state trajectories in different regions make different average degrees of turns on most timescales, on a universal timescale of ~25 ms, all state trajectories uniformly make ~100˚ turns on average, that are near orthogonal (i.e., near 90˚) but slightly conservative. This implies that successive state-trajectory changes on the timescale of ~25 ms are globally coordinated to be near-maximally unrelated, that is, near-minimally constrained, but slightly restoring.

How might time-averaged trajectory instability be globally maintained at ~100˚ on the timescale of ~25 ms? The timescale dependence of time-averaged trajectory-instability for each region depends on the exact spectral-amplitude profiles at its constituent sites. Nevertheless, there are some factors that make systematic contributions. For example, a Gaussian-random-walk type of EEG activity occurring at all constituent sites would yield a constant level of time-averaged trajectory instability at 90˚ regardless of timescale (making memory free orthogonal turns on all timescales on average).

Oscillatory activities occurring at any of the constituent sites would also make characteristic contributions. For example, if EEG at most sites oscillated at the same frequency, the state trajectory would move along an approximate elliptical path. The elliptical path would be planar only if EEG at all sites oscillated at the same frequency (with a non-singular distribution of phase). Presence of oscillations in other frequencies and the $1/f^b$ spectral background would distort the elliptical path in a variety of ways. Nevertheless, if oscillations at a specific frequency were dominant across sites relative to other contributions, the path would locally approximate a curved helix. In that case, average trajectory instability would be near 180˚ on the timescale of a half period (i.e., approximately reversing directions from half period to half period) and near 0˚ on the timescale of a period (i.e., approximately moving in the same direction from period to period). Oscillatory directional changes would again optimally impact the state trajectory on the timescale of 1.5 periods, and minimally impact it on the timescale of 2 periods, and so on, peaking at the odd multiples and dipping at the even multiples of the oscillatory half period. For real EEG data that include a broad range of oscillation frequencies, the impact of a specific oscillation frequency would diminish for the higher harmonics because the directional instability on longer timescales would be more strongly influenced by slower oscillatory components and slower-varying non-oscillatory components (including the random-walk type activity that partially underlies the $1/f^b$ spectral profile).

An example of an oscillatory contribution can be seen in the current results for the posterior region whose constituent sites yielded a substantial spectral peak in the $\alpha$-band (~10 Hz with 50 ms half-period) (the black curve in Fig 2). Based on our reasoning above, the $\alpha$-band contribution to the time-averaged trajectory instability should generate diminishing peaks on the timescales of 50 ms (a half-period), 150 ms (3 half-periods), etc., and diminishing troughs on the timescales of 100 ms (2 half-periods), 200 ms (4 half-periods), etc. This is what we observed (the black curve in Fig 3A). However, note that even the largest peak and trough on the half-period and one-period timescales only reached ~130˚ and ~75˚, respectively (rather than ~180˚ and ~0˚) because of the contributions of other oscillation frequencies as well as the

$1/f^b$ background (Fig 2), indicating that the overall posterior state trajectory was far from being periodic. As an alternative way to assess whether the obtained state trajectories were overall periodic, we examined the changes in trajectory direction as a function of temporal delay. We chose $\Delta t = {\sim}25$ ms (not exactly 25 ms due to the 512 Hz sampling rate) for the computation of interpolated trajectory directions $\Delta \overrightarrow{V_k}(t)$ because directional changes in the primary $\alpha$-band (${\sim}10$ Hz) oscillatory component would be optimally detected on the timescale of its quarter period (or 25 ms), sensitively detecting ${\sim}10$ Hz variations while effectively interpolating over higher-frequency fluctuations. The average angular difference indeed minimized at a delay of 100 ms (one complete cycle of a 10 Hz oscillation), consistent with the primary contributions of $\alpha$-band oscillations. However, whereas the minimum angular difference across a one-cycle delay should be 0˚ if the state trajectory was periodic, even for the posterior region in which the $\alpha$-band peak was most prominent (Fig 2), the minimum angular difference was 66.6˚ (*se* = 2.8˚), indicating that the state trajectories shifted by as much as 66.3˚ (not far from 90˚) on average across each $\alpha$ cycle. This confirms that the obtained state trajectories were far from being periodic.

Similar but less pronounced peaks (at 50 and 150 ms) and troughs (at 100 and 200 ms) were observed in the trajectory instability profiles for the central and left/right-lateral regions (the blue, orange, and green curves in Fig 3A), consistent with the fact that the $\alpha$-band peaks were less pronounced in the central and left/right-lateral regions (the blue, orange, and green curves in Fig 2) relative to the posterior region. The anterior instability profile was relatively flat on the timescales beyond ${\sim}15$ ms (the red curve in Fig 3A), consistent with the absence of any substantial spectral peaks (the red curve in Fig 2). Nevertheless, this relatively flat portion reflects oscillatory activities in a broad range of frequencies rather than a dominance by the random-walk type spectral background because the plateaued instability levels over ${\sim}15$–70 ms timescales were consistently above the 90˚ level expected by random walk (Fig 3A). The trajectory-instability profiles from all regions gradually converged toward the level of 90˚ on the longer timescales > ${\sim}150$ ms (Fig 3A), indicating that state trajectories on average make relatively unconstrained directional changes (with successive trajectory changes being maximally unrelated) on the timescales beyond ${\sim}150$ ms.

Given that trajectory instability profiles are sensitive to oscillatory activities and $1/f^b$ spectral baselines at constituent sites, the fact that the profiles from all five regions tightly converged at the level of ${\sim}100$˚ on the timescale of ${\sim}25$ ms (Fig 3A) may have functional relevance rather than being epiphenomenal. The convergence may suggest, for example, that intrinsic neural dynamics are globally calibrated in such a way that bottom-up sensory and top-down attentional/cognitive processes could exchange information in ${\sim}25$ ms packets at a matched level of nearly unconstrained but slightly conservative (${\sim}100$˚) state-trajectory flexibility. This matching may facilitate directionally balanced interfacing of bottom-up and top-down processes so that they contribute on an equal footing to construct internal representations that optimally incorporate both behavioral goals and environmental constraints. Because $\beta$-band activity (${\sim}20$ Hz, ${\sim}25$ ms half-period) would have the largest impact on state-trajectory instability on the convergent timescale of ${\sim}25$ ms, an efficient way to maintain this calibration may be to network-wide tune $\beta$-band activity to compensate for any cross-network variations in multi-frequency oscillatory and non-oscillatory activity.

While these inferences are highly speculative, the data provide some indirect support for the interpretation that the inter-region convergence of time-averaged trajectory instability at ${\sim}100$˚ at ${\sim}25$ ms may be actively maintained. If the convergence were actively maintained, time-averaged trajectory instability should be particularly stable on the critical timescale of ${\sim}25$ ms in all five regions, both across participants and over time (within each participant).

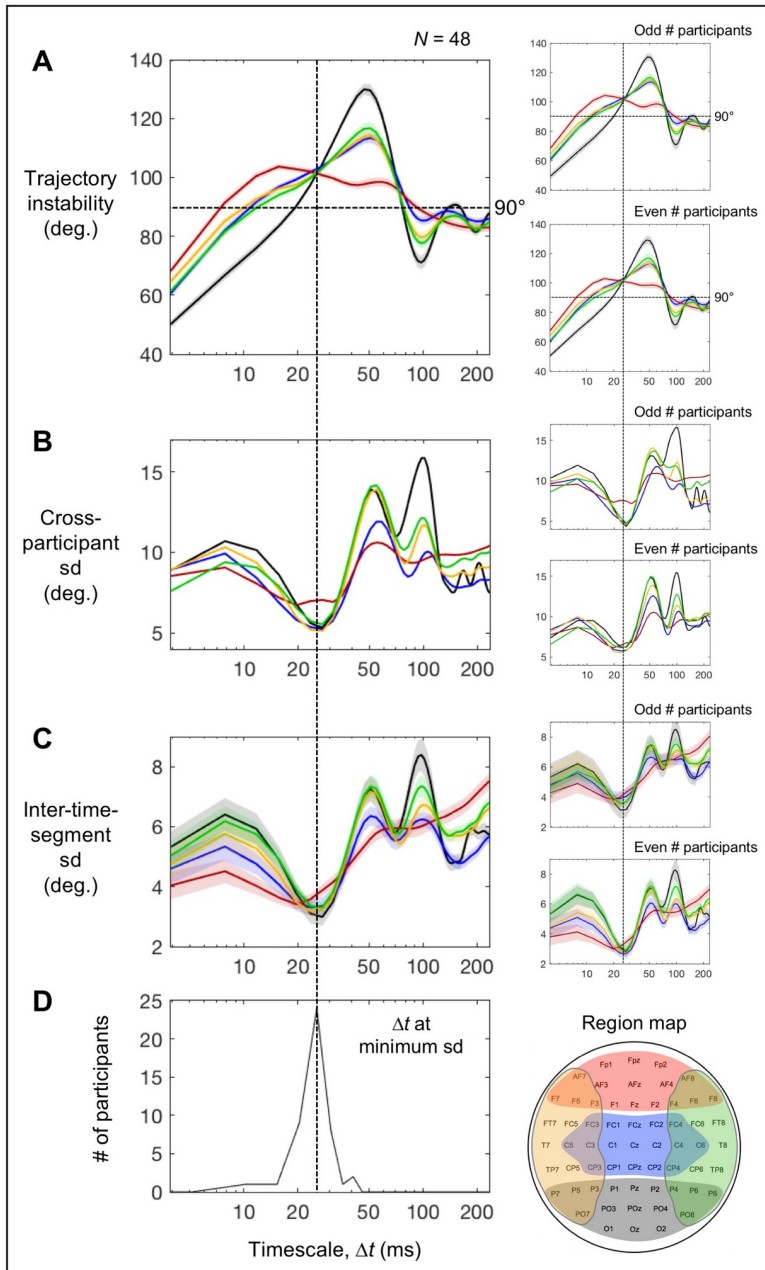

**Fig 7. The coincidence of the global convergence (across all five regions) of time-averaged trajectory instability and its minimum variability on the timescale of ~25 ms.** The five regions, posterior (black), central (blue), anterior (red), left-lateral (orange), and right-lateral (green), are color coded. **A.** Time-averaged trajectory instability as a function of timescale $\Delta t$ (the same as Fig 3A). **B.** Cross-participant standard deviation of time-averaged trajectory instability as a function of timescale $\Delta t$. **C.** Inter-time-segment standard deviation (trajectory instability averaged within each time segment) as a function of timescale $\Delta t$. **D.** Histogram of the timescale values that minimized the inter-time-segment standard deviation (computed based on the standard-deviation values averaged across all five regions). The vertical dashed lines show that time-averaged trajectory instability globally (across all regions) converged and achieved minimum variability (in terms of both cross-participant and inter-time-segment variability) on the timescale of ~25 ms. The shaded areas represent ±1 standard error of the mean. In **C**, standard errors have been adjusted to reflect the cross-participant consistency of timescale dependence independently of baseline individual differences (relevant to estimating the location of the global minimum) (Morey, 2008). For each plot, the reliability of the data patterns is demonstrated by the accompanying plots on the right showing virtually identical patterns obtained from the odd and even numbered participants.

Indeed, the cross-participant standard deviation of trajectory instability was minimized at the critical ~25 ms timescale in all five regions (Fig 7B) and the results were virtually identical for the odd and even numbered participants (the right panels in Fig 7B). To evaluate the temporal stability of time-averaged trajectory instability, we computed average trajectory instability over each consecutive 5-sec time segment, then evaluated its temporal variability as the standard deviation across all segments—inter-time-segment sd. This measure of temporal variability also minimized at the critical ~25 ms timescale in all five regions (though shifted to ~20 ms in the anterior region) (Fig 7C); the results were again virtually identical for the odd and even numbered participants (the right panels in Fig 7C). The histogram of the variability-minimizing timescale values (derived from the inter-time-segment sd profiles averaged across all regions) was sharply peaked at the critical ~25 ms timescale (Fig 7D), indicating that time-averaged trajectory instability was most temporally stable on the critical ~25 ms timescale for most participants.

Note that perturbations in any dimensions orthogonal to the plane of a trajectory turn would shift the degree of the turn toward 90˚, and the shift would be larger for trajectory turns at angles more deviated from 90˚ (with no shifts for 90˚ turns). Thus, random perturbations would generate greater variability when average turns (reflected in time-averaged trajectory instability) are at angles more deviated from 90˚. One can confirm this prediction for time-scales longer than the critical ~25 ms timescale. For all regions, when the time-averaged trajectory instability substantially deviated from 90˚ (e.g., on ~50 ms and ~100 ms timescales for the posterior, central, and left/right-lateral regions; Fig 7A), both the cross-participant and inter-time-segment sd's tended to be at local maxima (Fig 7B and 7C), whereas when the time-averaged trajectory instability crossed (or approached) the 90˚ level (e.g., on ~75 ms and ~160 ms timescales for the posterior region; Fig 7A), both the cross-participant and inter-time-segment sd's tended to be at local minima (Fig 7B and 7C). Remarkably, despite the fact that the time-averaged trajectory instability was substantially higher than 90˚ on the critical ~25 ms time-scale (Fig 7A), both the across-participant and inter-time-segment sd's hit their global minima (Fig 7B and 7C) for all regions. This supports the hypothesis that the global convergence of time-averaged trajectory instability (across all five regions) at ~100˚ on the critical timescale of ~25 ms may be actively maintained. Thus, future investigations into the potential mechanisms and functional relevance of this convergence are warranted.

In contrast to the feature-rich profiles of time-averaged trajectory instability, time-averaged trajectory speed largely yielded monotonically decreasing profiles as a function of timescale for all five regions (Fig 3F). Gaussian-random-walk type EEG activity occurring at all sites would yield a power-function profile of time-averaged trajectory speed, which would be a negatively sloped line in the log-log plot, with the intercept proportional to the temporal standard deviation of EEG. An oscillatory activity would contribute maximal directional changes twice per oscillation period. Thus, on timescales sufficiently shorter than a half-period, the path-segments $\Delta \overrightarrow{V_k}(t)$ linearly interpolating over $\Delta t$ (used in the current analyses; Fig 1) would relatively closely track the periodic directional changes. However, for timescales surpassing a half-period, the path segments linearly interpolating over $\Delta t$ would substantially smooth over the periodic directional changes, yielding shorter interpolated trajectories with slower speeds. Thus, an oscillatory component would contribute a relatively constant average speed up to the timescale of about a half-period, but the contribution would rapidly decrease on longer time-scales. As a result, an oscillatory component would flatten the negatively sloped linear profile of time-averaged trajectory speed (reflecting the $1/f^b$ spectral-amplitude profile in log-log) for timescales up to about a half-period, thereby making the profile upwardly bowed with a broad peak trailing the timescale of a half period. This reasoning is consistent with the fact that the

posterior state trajectory with its prominent $\alpha$-band peak (the black curve in Fig 2) yielded a substantially bowed trajectory-speed profile with the curvature broadly peaking on the time-scale a little longer than ~50 ms (the black curve in Fig 3F). The degree of bowing is less for the central and left/right-lateral regions (the blue, orange, and green curves in Fig 3F) with less prominent $\alpha$-band peaks (the blue, orange, and green curves in Fig 2). The anterior region with no particularly prominent spectral peaks yielded a relatively linear (in log-log) time-averaged trajectory-speed profile (the red curve in Fig 3F).

We note that the notch filter at 60 Hz used to remove line noise (see Materials and Methods) attenuated oscillatory components that would have strongly influenced trajectory instability and speed on ~10 ms and shorter timescales (note that the half period of 60 Hz is ~8 ms). The filter thus caused the general flattening of the time-averaged trajectory-speed profiles (Fig 3F) as well as the general falling of the time-averaged trajectory-instability profiles (Fig 3A) on ~10 ms and shorter timescales.

## Influences of phase relations on EEG state-trajectory instability and speed

While the discussion so far has focused on the influences of spectral-amplitude profiles on EEG-state trajectory instability and speed, our results suggest that they are also influenced by phase relations. We used two types of phase scrambling to infer the contributions of cross-site within-frequency and cross-frequency phase relations. Comparisons with the full-phase-scrambled controls (EEG from each site independently phase-scrambled) elucidated the joint contributions of cross-site within-frequency and cross-frequency phase relations to trajectory instability and speed, whereas comparisons with the cross-frequency phase-scrambled controls (EEG from all sites phase-scrambled with the same randomization seed thus preserving cross-site within-frequency phase relations) isolated the contributions of cross-frequency phase relations.

**Influences of phase relations on trajectory instability.** We first consider the potential impact of cross-site within-frequency phase relations on state-trajectory instability, and compare them with the results. At one extreme, if oscillatory activities in a specific frequency band were all phase aligned (i.e., 0° phase lagged) across sites, they would contribute an approximately oscillatory component to a state trajectory. At another extreme, if their phases were minimally aligned across sites, that is, if the phases were uniformly distributed between 0° and 180° across sites, the oscillatory activities would contribute an approximately circular component to a state trajectory. Strictly speaking, the contribution would be circular only if oscillation amplitudes were equal across sites. Uneven amplitudes would make the contribution elliptical. Given a set of uneven amplitudes, the contribution would have the smallest aspect ratio (i.e., closest to being circular) when the phases of large-amplitude oscillations were evenly distributed between 0° and 180°. An intermediate level of cross-site phase alignment would contribute an approximately elliptical component with a higher (narrower) aspect ratio when the phases are more tightly aligned across sites. Note that phase scrambling would reduce the aspect ratio of these elliptical contributions but would not make them circular because a randomized phase distribution would necessarily deviate from a uniform distribution. As expected, when we phase scrambled artificial data consisting of single-frequency, equal-amplitude oscillatory sources with evenly distributed phases across the sources, the originally circular trajectory became elliptical.

Because trajectory instability is approximately a timescale dependent measure of trajectory curvature, a narrow elliptical component would generate large temporal variability within each oscillatory cycle because directional changes would be large over the two high-curvature portions and small over the two low-curvature portions. Because each of the high- and low-

curvature portions spans about a half cycle, an elliptical component would particularly elevate trajectory-instability variability measured on the timescale of a quarter period (sharply changing directions over the half period containing the high curvature and largely moving in the same direction over the half period containing the low curvature). This cyclic variation in trajectory instability would become less pronounced for elliptical components with lower aspect ratios generated by less tight cross-site phase alignment because the high and low curvatures would be less different. Thus, a tighter cross-site within-frequency phase alignment of oscillatory activities should increase trajectory-instability variability especially on the quarter-period timescale. In contrast, time-averaged trajectory instability is not expected to be much affected by cross-site within-frequency phase alignment because the average curvature of an elliptical path would be relatively independent of its aspect ratio. We confirmed this reasoning by simulating the waveform at each site as a sinewave oscillating at $f$ Hz and varying the tightness of the phase distribution across 17 sites (because our EEG analyses included 17 sites per region). As predicted, tighter cross-site phase alignment yielded greater trajectory-instability variability, especially on the timescale of the quarter oscillatory period (and also on the timescales of its harmonics if no $1/f$ noise was included), with little effect on time-averaged trajectory instability. Our simulations further demonstrated that cross-site phase synchronization over a broad range of frequencies elevated trajectory-instability variability across a broad range of timescales with little effect on time-averaged trajectory instability. Specifically, we simulated the waveform at each site as either a Gaussian noise (with a flat spectral-amplitude profile) or a Gaussian random-walk noise (with a $1/f$ spectral-amplitude profile typically observed in human EEG; e.g., Fig 2), thus simulating phase-synchronized broadband activity across sites. In each case, the trajectory-instability variability for the broadband cross-site phase-synchronized noise was substantially elevated in a largely timescale-invariant manner relative to the full phase-scrambled control that randomized cross-site within-frequency phase relations, whereas the time-averaged trajectory instability minimally deviated from the control.

The resting EEG data across all regions mirrored these simulation results (largely timescale-invariant elevations in trajectory-instability variability, Fig 4C, with minimal, less than ±2˚, effects on time-averaged trajectory instability, Fig 4A, relative to the full phase-scrambled controls), suggesting that intrinsic neural dynamics equivalently facilitate phase-synchronization across a broad range of frequencies. The regional differences in the elevation of trajectory-instability variability (Fig 4C) suggest that cross-site broadband synchronization was most facilitated in the posterior region, slightly less facilitated in the anterior and left/right-lateral regions, and least facilitated in the central region. These differences may reflect local computational needs that may require different degrees of broadband phase synchronization.

Interestingly, trajectory-instability variability was not elevated relative to the cross-frequency phase-scrambled controls (Fig 5C). This indicates that any coordination of within-site cross-frequency interactions is such that it does not substantially influence trajectory-instability variability. The potential cross-frequency mechanisms underlying the small but consistent reductions in trajectory-instability variability in specific ranges of timescales in the posterior region (the black curve in Fig 5C) are unclear.

**Influences of phase relations on trajectory speed.** A theoretical consideration suggests that the coordination of cross-site within-frequency phase synchronization and the coordination of within-site cross-frequency interactions may both influence time-averaged trajectory speed and its temporal variability in opposite directions. First, let us consider a case where each of the $k$ sites made a brief, lasting $\Delta T$, voltage change of $\Delta v$ within the time period of $N \times \Delta T$. For simplicity, we assume that $N = k$ (i.e., as many brief time periods of $\Delta T$ as the number of sites). At one extreme, if these changes across the $k$ sites occurred completely

desynchronously (i.e., each site making the brief change in a separate $\Delta T$ interval), the trajectory speed at each $\Delta T$ would be constant at $\frac{\Delta v}{\Delta T}$ (ignoring directional changes for brevity), so that the time-averaged trajectory speed would be, $\frac{\sum_{i=1}^{i=k} \frac{\Delta v}{\Delta T}}{k} = \frac{\Delta v}{\Delta T}$, and the temporal variability in trajectory speed would be zero. At the other extreme, if the changes across the $k$ sites all occurred synchronously within a single $\Delta T$ interval, the trajectory speed would be

$\frac{\sqrt{\sum_{i=1}^{i=k} \Delta v^2}}{\Delta T} = \frac{\Delta v}{\Delta T} \times \sqrt{k}$ (due to vector summation) within that interval and zero within all other intervals, so that the time-averaged trajectory speed would be $\frac{\frac{\Delta v}{\Delta T} \times \sqrt{k}}{k} = \frac{\Delta v}{\Delta T} \times \frac{1}{\sqrt{k}}$; the temporal variability in trajectory speed (in standard deviation) would be $\frac{\Delta v}{\Delta T} \times \sqrt{k}$.

Thus, increased cross-site synchronization of EEG waveforms should both reduce time-averaged trajectory speed (from $\frac{\Delta v}{\Delta T}$ to $\frac{\Delta v}{\Delta T} \times \frac{1}{\sqrt{k}}$ in this example) and increase its temporal variability (from 0 to $\frac{\Delta v}{\Delta T} \times \sqrt{k}$ in this example). We can also reason that these synchronization effects would be enhanced when EEG changes at individual sites were sharper (i.e., larger $\frac{\Delta v}{\Delta T}$) and more intermittent (enhancing the impact of synchronization; cf. if EEG changes were constant at each site, cross-site synchronization would have no impact).

Given these considerations, increases in cross-site synchronization and the coordination of within-site cross-frequency phase relations that generates sharper and intermittent EEG changes may independently contribute to reductions in time-averaged trajectory speed and increases in its variability relative to the full phase-scrambled controls (Fig 4B and 4D) and the smaller effects relative to the cross-frequency phase-scrambled controls (Fig 5B and 5D). The fact that the large effects relative to the full phase-scrambled controls were largely timescale invariant implies the coordination of broadband cross-site phase synchronization. In partial support of the unique effects of cross-site within-frequency synchronization, in the simulation experiments discussed above, the phase-synchronized broadband noise (Gaussian or Gaussian random-walk noise) with random cross-frequency phase relations generated largely timescale-invariant reductions in time-averaged trajectory speed and increases in its variability relative to only the full phase-scrambled controls (with little effect of cross-frequency phase scrambling).

The fact that the smaller effects relative to the cross-frequency phase-scrambled controls were more tightly timescale and region invariant (Fig 5B and 5D) suggests that cross-frequency phase relations are globally coordinated in such a way that EEG waveforms at each site include sharp and intermittent changes across a broad range of timescales. This may be analogous to the way in which different spatial-frequency components in natural visual scenes are phase aligned to generate sharp, high-contrast, and sparse object borders across a broad range of spatial scales (partly due to changes in viewing distance).

If cross-site within-frequency synchronization and within-site cross-frequency phase coordination caused both reductions in time-averaged trajectory speed and increases in its variability, the reductions and increases should be closely associated over time. For example, during a period when time-averaged trajectory speed is strongly reduced its variability should be strongly increased. To support the role of cross-site within-frequency synchronization, we should see a strong temporal association between average-speed reductions and speed-variability increases measured relative to the full phase-scrambled controls beyond the cross-frequency phase-scrambled controls. To support the role of within-site cross-frequency coordination, we should see a strong temporal association between average-speed reductions and speed-variability increases measured relative to the cross-frequency phase-scrambled controls. Because the average-speed reductions and speed-variability increases were largely

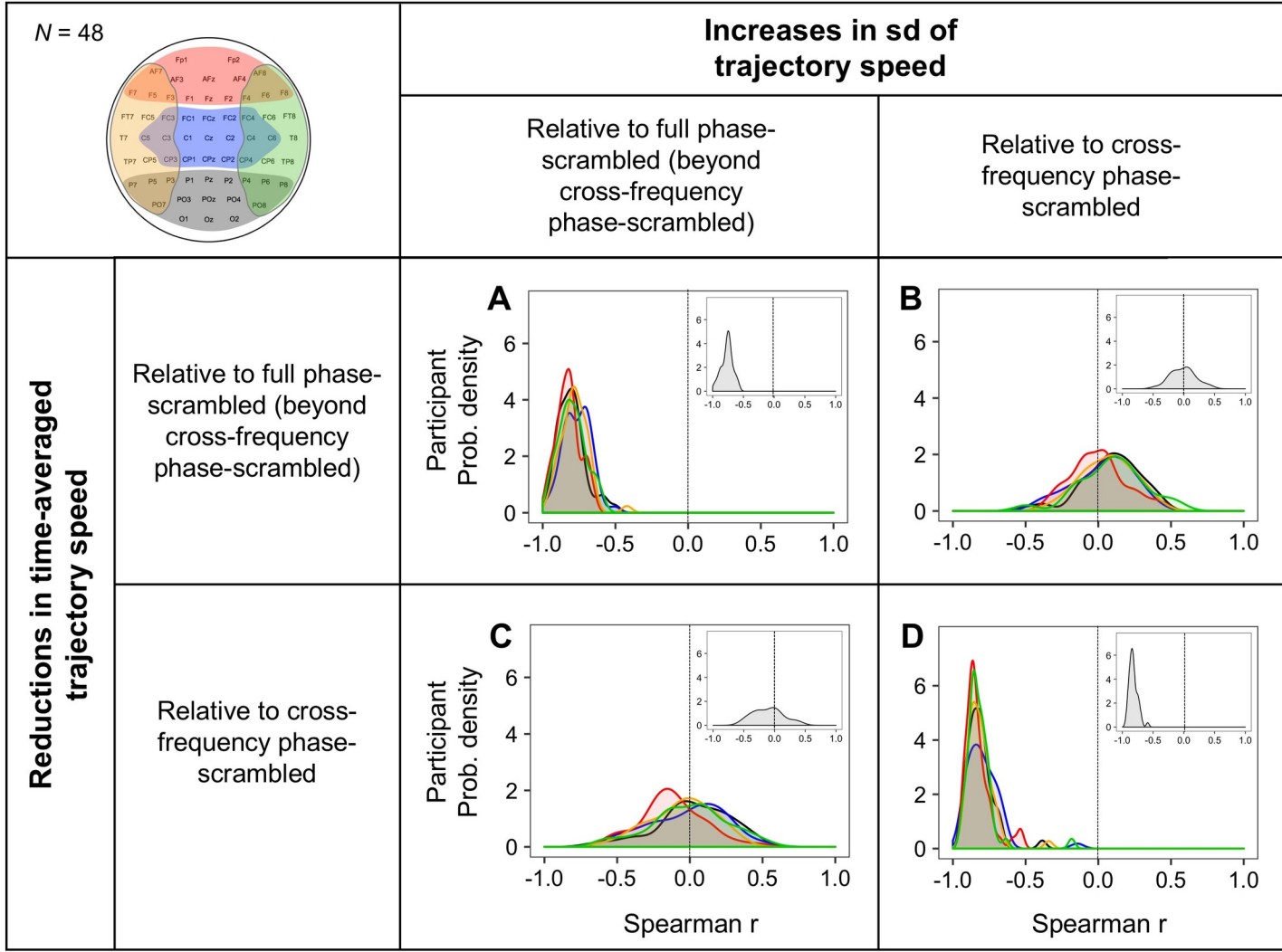

**Fig 8. Temporal associations (across 5-sec time segments) between reductions in time-averaged trajectory speed and increases in its standard deviation (sd).** Each variable was measured relative to the full phase-scrambled controls (beyond the cross-frequency phase-scrambled controls) to reflect the effects of cross-site within-frequency phase relations, and relative to the cross-frequency phase-scrambled controls to reflect the effects of cross-frequency phase relations. Temporal associations are organized in a 2-by-2 design where the two hypothetical underlying mechanisms, the coordination of cross-site within-frequency phase relations and the coordination of cross-frequency phase relations, are matched on the diagonal (**A** and **D**) and crossed on the off diagonal (**B** and **C**). Each panel shows the cross-participant distributions (as interpolated probability density functions) of Spearman r values for the five regions (posterior: black, central: blue, anterior: red, left-lateral: orange, and right-lateral: green). The insets show the corresponding distributions for EEG-state trajectories for the global non-adjacent site configuration (see Fig 10).

timescale invariant (Figs 4B, 4D, 5B and 5D), we averaged the values across timescales (though we verified that the temporal associations were equivalent across timescales). As we computed all statistics (i.e., temporal averages and standard deviations of trajectory instability and speed as well as their phase-scrambled controls) within each consecutive 5-sec time segment (see Materials and Methods), we computed the temporal associations between average-speed reductions and speed-variability increases across time segments per region per participant. We used Spearman *r* to minimize the effects of potential outliers.

In support of our predictions, we obtained strong negative temporal associations between average speed and speed variability in all five regions from all participants ($r < -0.5$ for nearly all participants) both relative to full phase-scrambled controls (beyond cross-frequency phase-

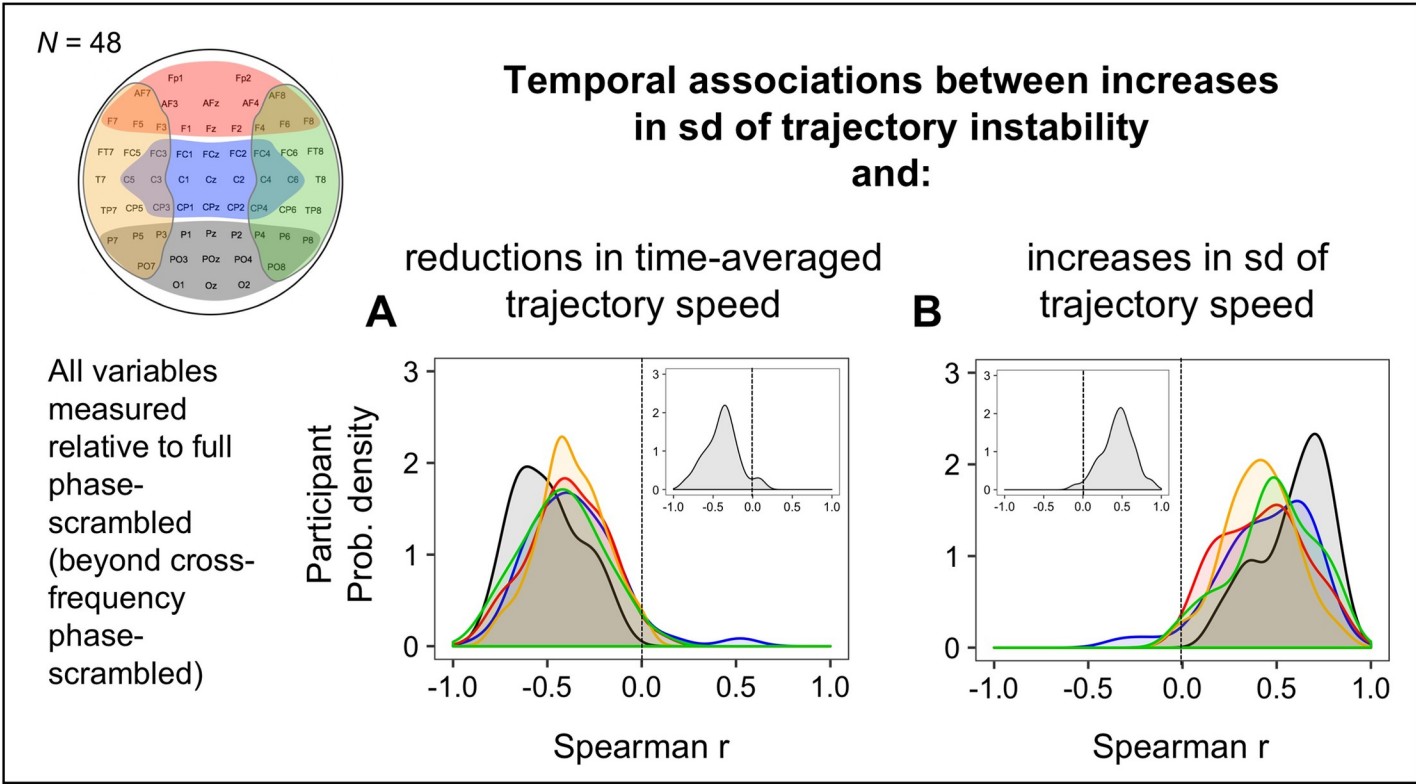

**Fig 9. Temporal associations (across 5-sec time segments) between increases in the standard deviation (sd) of trajectory instability and reductions in time-averaged trajectory speed and increases in the sd of trajectory speed.** All variables were measured relative to the full phase-scrambled controls (beyond the cross-frequency phase-scrambled controls) to reflect the effects of cross-site within-frequency phase relations. **A.** Temporal associations between increases in trajectory-instability sd and reductions in time-averaged trajectory speed. **B.** Temporal associations between increases in trajectory-instability sd and increases in trajectory-speed sd. Each panel shows the cross-participant distributions (as interpolated probability density functions) of Spearman r values for the five regions (posterior: black, central: blue, anterior: red, left-lateral: orange, and right-lateral: green). The insets show the corresponding distributions for EEG-state trajectories for the global non-adjacent site configuration (see Fig 10).

scrambled controls) (Fig 8A) and relative to cross-frequency phase-scrambled controls (Fig 8D). To confirm the specificity of these associations, we obtained no consistent temporal associations when the types of phase-scrambled controls were crossed between the measures of average-speed reductions and speed-variability increases (Fig 8B and 8C). Thus, both cross-site within-frequency synchronization and within-site cross-frequency phase coordination (e.g., potentially generating sharp and intermittent changes) globally reduce time-averaged trajectory speed and increase its variability, but these mechanisms operate independently.

Earlier we reasoned (based on the EEG and simulation results) that cross-site within-frequency synchronization also increases the variability in trajectory instability. If so, increases in trajectory-instability variability should be temporally associated with the aspects of average-speed reductions and speed-variability increases attributable to the coordination of cross-site within-frequency phase synchronization. Indeed, when these variables were measured relative to the full phase-scrambled controls (beyond the cross-frequency phase-scrambled controls), increases in trajectory-instability variability were negatively associated with average-speed reductions ($r < 0$ for nearly all participants; Fig 9A) and positively associated with speed-variability increases ($r > 0$ for nearly all participants; Fig 9B) in all five regions.

Thus, the temporal association results confirm that the coordination of cross-site within-frequency synchronization and the coordination of within-site cross-frequency phase relations

independently contribute to reducing time-averaged trajectory speed and increasing its variability, with the coordination of cross-site within-frequency synchronization additionally contributing to increasing the variability in trajectory instability.

**Influences of phase relations on the temporal correlation between trajectory instability and speed.** Overall, we obtained positive temporal correlations between trajectory instability and speed relative to both the full phase-scrambled and cross-frequency phase-scrambled controls, suggesting that cross-site within-frequency and cross-frequency phase relations are both adjusted in such a way that faster trajectories are less directionally stable and slower trajectories are more directionally stable. Note that an oscillatory component would generally contribute a negative correlation because the magnitudes of the 1st and 2nd derivatives, closely related to our measures of trajectory speed and instability, are negatively correlated in sinusoidal oscillations. Thus, the generally positive temporal correlations (relative to both types of phase-scrambled controls) suggest that cross-site within-frequency and cross-frequency phase relations are both adjusted in such a way that EEG-state trajectories are less oscillatory in all regions. We additionally obtained some characteristic regional differences (Figs 4E and 5E), but further research is necessary to elucidate the mechanisms underlying these effects.

## Consideration of volume conduction effects

The results indicative of cross-site phase synchronization in EEG (oscillatory or non-oscillatory) could have been influenced by spurious contributions of volume conduction. The potentially affected results include increases in the temporal variability in trajectory instability as well as reductions in time-averaged trajectory speed and increases in its variability, all of which at least partially implicated cross-site within-frequency synchronization. For a standard 64-site configuration, volume conduction effects after applying the surface-Laplacian transform are minimal beyond the distance between adjacent sites (e.g., [35]). We thus re-examined these effects for state trajectories of a global array comprising 16 non-adjacent sites evenly sampled from the whole head (see the map in Fig 10). If spurious volume conduction effects played substantial roles in the original analyses, the effects should minimize in this re-analysis.

All effects replicated. The temporal variability in trajectory instability was consistently elevated relative to the full phase-scrambled control across all timescales, but was largely at the same level as the cross-frequency phase-scrambled control (Fig 10A), replicating the results obtained for the five regions (Figs 4C and 5C). The time-averaged trajectory speed was consistently reduced relative to both the full and cross-frequency phase-scrambled controls across all timescales, and consistently more reduced relative to the full than cross-frequency phase scrambled control across all timescales (Fig 10B), replicating the results obtained for the five regions (Figs 4B and 5B). The temporal variability in trajectory speed was consistently elevated relative to both full and cross-frequency phase-scrambled controls across all timescales, and consistently more elevated relative to the full than cross-frequency phase scrambled control across all timescales (Fig 10C), replicating the results obtained for the five regions (Figs 4D and 5D). All these results were virtually identical for the odd and even numbered participants (the right panels accompanying all main panels in Fig 10). We also replicated all temporal-association effects involving reductions in time-averaged trajectory speed, increases the variability in trajectory speed, and increases in the variability in trajectory instability (insets in Figs 8 and 9). Thus, our conclusions on the effects of phase relations are unlikely to be explained by volume conduction.

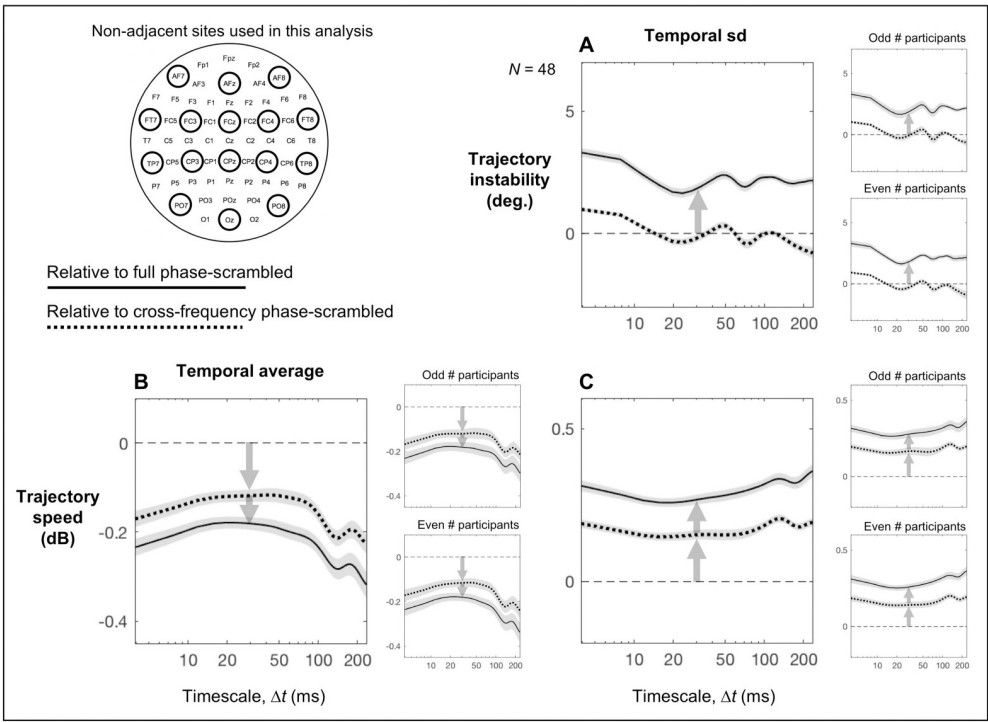

**Fig 10. An evaluation of potential contributions of volume conduction to the effects attributable to cross-site phase synchronization.** Because volume conduction effects are negligible beyond adjacent scalp sites on a standard 64-site array after applying the surface-Laplacian transform (e.g., [35]), we re-evaluated the effects that may be susceptible to volume conduction, (A) increases in the temporal variability in trajectory instability, (B) reductions in time-averaged trajectory speed and (C) increases in its temporal variability, for state trajectories of a global array comprising 16 non-adjacent sites (see upper left). The effects should minimize if volume conduction played substantial roles in the original analyses. The effects relative to the full phase-scrambled controls are shown with solid lines and those relative to the cross-frequency phase-scrambled controls are shown with dotted lines. The shaded areas represent ±1 standard error of the mean. **A.** The temporal variability (measured as temporal standard deviation) in trajectory instability as a function of timescale Δt relative to the phase-scrambled controls. It was consistently elevated relative to the full phase-scrambled control across all timescales, but was largely at the level of the cross-frequency phase-scrambled control, replicating the original analyses presented in Figs 4C and 5C. **B.** Time-averaged trajectory speed as a function of timescale Δt relative to the phase-scrambled controls. It was consistently reduced relative to both the full and cross-frequency phase-scrambled controls across all timescales, and consistently more reduced relative to the full than cross-frequency phase-scrambled control across all timescales. These results replicate the original anlayses presented in Figs 4B and 5B. **C.** The temporal variability (measured as temporal standard deviation) in trajectory speed as a function of timescale Δt relative to the phase-scrambled controls. It was consistently elevated relative to both the full and cross-frequency phase-scrambled controls across all timescales, and consistently more elevated relative to the full than cross-frequency phase-scrambled control across all timescales. These results replicate the original analyses presented in Figs 4D and 5D. For each plot, the reliability of the data patterns is demonstrated by the accompanying plots on the right showing virtually identical patterns obtained from the odd and even numbered participants.

## Concluding remarks

A major goal of the current study was to elucidate potential macroscopic rules governing global neural dynamics by examining EEG spatiotemporal state trajectories.

A first potential rule is the tight inter-region convergence of time-averaged state-trajectory instability at the level of ~100˚ on the timescale of ~25 ms (Fig 3A–3D). Both individual differences and temporal variability in trajectory instability minimized on the critical timescale of ~25 ms (Fig 7B and 7D) providing indirect evidence suggesting that the convergence is actively maintained. Thus, it is possible that spectral-amplitude profiles (e.g., Fig 2) are globally adjusted to maintain (on average) EEG spatiotemporal dynamics at a nearly directionally

unconstrained but slightly conservative level across all regions on a universal timescale of ~25 ms. We speculated that this convergence may establish a global "conduit" for flexible (nearly unconstrained) information exchanges in ~25 ms units.

A second potential rule is that the broadband coordination of cross-site phase synchronization globally (across all regions) generates largely timescale-invariant increases in the temporal variability in trajectory instability, reductions in time-averaged trajectory speed, and increases in its temporal variability. The same cross-site phase synchronization mechanisms appear to generate all these effects based on theoretical considerations, simulation results, and the fact that the effects were closely temporally associated with one another (Figs 8A and 9A and 9B).

A third potential rule is that the coordination of within-site cross-frequency phase relations, that likely generates sharper and more intermittent changes that are self-similar across timescales, globally (across all regions) generates largely timescale-invariant reductions in time-averaged trajectory speed and increases in its temporal variability. The same within-site cross-frequency coordination mechanisms appear to generate both effects as they were closely temporally associated with each other (Fig 8D).

In comparing the rules two and three, while both the broadband coordination of cross-site phase synchronization and the coordination of within-site cross-frequency phase relations generate largely timescale-invariant reductions in time-averaged trajectory speed and increases in its temporal variability, the diagonal pattern of temporal association results suggest that these contributions are independent (Fig 8).

A fourth potential rule is that both the broadband coordination of cross-site phase synchronization and the coordination of within-site cross-frequency phase relations generate overall positive temporal correlations between trajectory instability and speed in all regions, thus globally nudging EEG state trajectories away from being oscillatory.

A deeper understanding of these potential global rules governing intrinsic neural dynamics will depend on future research investigating how the behaviors of the spectral-amplitude dependent and phase-relation dependent aspects of state-trajectory instability and speed change while people engage in different behavioral tasks. Some of the global rules of intrinsic dynamics may remain task invariant, implying homeostatic processes; disruptions of such rules may result in behavioral dysfunctions. Other rules may characteristically change according to cognitive requirements and/or sensory interactions, elucidating their potential functional relevance. Some preliminary results are suggestive of these distinctions. For example, the inter-region convergence of time-averaged trajectory instability at 100° at ~25 ms (obtained while participants rested with their eyes closed in the current study) appears to be maintained during viewing of a nature video as well as when people view and rate flickering displays for aesthetic preference. In contrast, the effects of phase relations on state-trajectory instability and speed appear to change in a task and stimulus dependent manner (Menceloglu, Grabowecky, & Suzuki, unpublished results).

Most EEG research has focused on elucidating the roles of distinct neural sources or networks of sources in behavioral functions. Multi-electrode EEG signals are typically decomposed into estimated neural sources and then time-frequency decomposed so that functional networks can be identified based on spectral amplitude and phase associations. Even when spatiotemporal EEG trajectories are considered (as in the current study) they are typically dimensionally reduced based on linear associations (e.g., using a principal-component analysis) to find a relatively small number of functionally relevant components. Behaviors of the identified functional networks and components are examined in relation to performing various behavioral tasks. These approaches based on identifying behaviorally relevant networks have generated much knowledge in terms of neural correlates of behavior. The prevalence of this approach is evident in the literature (see Introduction) and contents of advanced EEG

analysis textbooks (e.g., [35]). Here, we took an alternative approach of examining basic features (approximately the 1st and 2nd derivatives) of EEG spatiotemporal trajectories and identified simple rules that appear to govern intrinsic neural dynamics while people rested with their eyes closed. An effort to understand the mechanisms that maintain these simple macroscopic rules may help to advance our understanding of the mechanisms that maintain the integrity of global neural dynamics.

## Author Contributions

**Conceptualization:** Satoru Suzuki.

**Data curation:** Melisa Menceloglu.

**Formal analysis:** Melisa Menceloglu, Satoru Suzuki.

**Funding acquisition:** Melisa Menceloglu.

**Investigation:** Melisa Menceloglu, Marcia Grabowecky, Satoru Suzuki.

**Methodology:** Melisa Menceloglu, Satoru Suzuki.

**Project administration:** Melisa Menceloglu, Marcia Grabowecky.

**Resources:** Marcia Grabowecky, Satoru Suzuki.

**Software:** Melisa Menceloglu, Satoru Suzuki.

**Supervision:** Marcia Grabowecky, Satoru Suzuki.

**Validation:** Melisa Menceloglu, Satoru Suzuki.

**Visualization:** Melisa Menceloglu, Satoru Suzuki.

**Writing – original draft:** Satoru Suzuki.

**Writing – review & editing:** Melisa Menceloglu, Marcia Grabowecky, Satoru Suzuki.

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
