## [Decision Letter · Decision Letter 0]

26 Mar 2020

PONE-D-19-30275

EEG state-trajectory instability and speed reveal global principles of intrinsic spatiotemporal neural dynamics

PLOS ONE

Dear Suzuki,

Thank you for submitting your manuscript to PLOS ONE. After careful consideration, we feel that it has merit but does not fully meet PLOS ONE’s publication criteria as it currently stands. Therefore, we invite you to submit a revised version of the manuscript that addresses the points raised during the review process.

In particular, one reviewer raised substantial concerns about some part of the preprocessing pipeline  and the impact it has on the results reported in the paper. 

We would appreciate receiving your revised manuscript by May 10 2020 11:59PM. To enhance the reproducibility of your results, we recommend that if applicable you deposit your laboratory protocols in protocols.io, where a protocol can be assigned its own identifier (DOI) such that it can be cited independently in the future. For instructions see: http://journals.plos.org/plosone/s/submission-guidelines#loc-laboratory-protocols

We look forward to receiving your revised manuscript.

Kind regards,

Giovanni Petri, Ph.D.

Academic Editor

PLOS ONE

Journal Requirements:

3. Please amend the manuscript submission data (via Edit Submission) to include author Melisa Menceloglu and Marcia Grabowecky.

Reviewers' comments:

Reviewer's Responses to Questions

**Comments to the Author**

1. Is the manuscript technically sound, and do the data support the conclusions?

Reviewer #1: Yes

Reviewer #2: Partly

2. Has the statistical analysis been performed appropriately and rigorously? 

Reviewer #1: Yes

Reviewer #2: N/A

3. Have the authors made all data underlying the findings in their manuscript fully available?

Reviewer #1: Yes

Reviewer #2: Yes

4. Is the manuscript presented in an intelligible fashion and written in standard English?

Reviewer #1: Yes

Reviewer #2: No

5. Review Comments to the Author

Reviewer #1: Manuscript summary:

The authors examined scalp EEG during waking quiescence in healthy adult humans. They characterized large-scale patterns EEG current source density using a pair of novel metrics, trajectory instability and speed, which describe the evolution of high-dimensional EEG states over time. Findings include that these quantities vary topographically across the head and when computed at differing time-scales and that this likely reflects topographical variation in CSD amplitude spectra. Comparison to phase-scrambled data showed that within- and across-frequency phase alignment affected trajectory instability variably substantially, but had modest effect when integrated over time. Synchronization results in time averaged reductions in trajectory speed, while increasing variability similarly to instability. In addition, these metrics are relatively stable for the (admittedly brief) 5-minute duration of the experiment. Notably, trajectory instability was found to converge at 100º at a timescale of ~25ms across the entire scalp, a phenomena warranting further investigation. The authors' thorough controls, including the use of simulated data and the accounting of volume conduction effects is commendable. As is their contribution to the sometimes underappreciated observational sciences.

Minor concerns:

1. In Figure 1C “Unstable” is misspelled “Untable”.

2. If the units of Vk are uV/mm^2, shouldn’t the units of trajectory speed be uV/mm^2·s rather than a.u.? In general, more explicit specification of units would be welcome.

3. Subject demographics are listed as male, female, and non-binary. However, male and female refer to sex, while non-binary refers to gender-identity. Recommended practice[1] would be to list male, female, and intersex, or men, women, and non-binary (or both); as appropriate.

4. The precise instructions given to participants before resting-state experiments is a matter of some debate and may influence brain activity[2]. For these reasons, and to facilitate reproducibility, could you please include the instructions given in you recording procedures?

5. Footnotes are not permitted in PLOS publications. Please incorporate them into the text.

References:

1. Clayton JA, Tannenbaum C. Reporting sex, gender, or both in clinical research?. Jama. 2016 Nov 8;316(18):1863-4.

2. Kawagoe T, Onoda K, Yamaguchi S. Different pre‐scanning instructions induce distinct psychological and resting brain states during functional magnetic resonance imaging. European Journal of Neuroscience. 2018 Jan;47(1):77-82.

Reviewer #2: The authors examined EEG resting state activity in healthy subjects aiming at the identification of quantitative principles governing large-scale neural dynamics. To do so, they divided the overall number of electrodes into three sub-networks corresponding to areas responsible for different cognitive functions. The paper is a methodological study - within the network science framework – aiming at the identification of relevant global signatures of the EEG signal.

Unfortunately, it is not reported whether the practice of subdividing the EEG resting state signal within three subnetworks is novel, and at the same time the authors do not clarify what they mean by global signature of neural dynamics.

The general feeling is that a comparison with the state of the art of the literature in some of the domains is lacking. Arguments that should justify the choices of the authors are missing. For comparison, some examples concerning cross-frequency coupling and mind-wondering (https://www.sciencedirect.com/science/article/abs/pii/S1364661310002068

works from Smallwood et al.)

The authors did summarize the main research question, however it is not clear how their key findings apply, i.e. which are the main applications of the authors’ research.

The authors should specify what they mean by quantitative principles and, whether they refer to the quantities in page 4, they should provide clear examples about how informative are those quantities. Furthermore, extracting features from the whole scalp EEG signal is a practice in the entire BCI literature. The whole literature is ignored, while it provides a real-time analysis of EEG features comparable with those the authors aim at defining in this article. If the authors aim at extracting intrinsic dynamics during spontaneous thoughts, it is not clear why do they subdivide the electrodes within 3 regions instead of looking at the whole brain connectivity. This is something that should be clarified or shifted in the methods section.

Methods section:

The authors should clarify how artifacts have been removed. Authors should justify according to the state of the art why did they apply the Laplacian transform (https://www.ncbi.nlm.nih.gov/pubmed/18347966), instead of other forms of correction for the references.

The authors should clarify why the signal was segmented differently depending on the participant

• 0.5 Hz is a frequency reasonable for resting state study. The claim that 5s is an amount of time which would allow comparison with cognitive tasks is not entirely true, as when it comes for visual task every trial last up to 500ms. The authors should therefore clarify with what type of cognitive tasks they are comparing their results to.

• The authors should also comment on the settings of the filters of the hardware system used (Biosemi)

• The authors should clarify why they divided in three relevant networks: if the practice is new, then they should state why it’s relevant – comparing with previous literature. If the practice is not new, then they should cite other studies that did the same.

Results and Discussion:

The discussion is not structured systematically, according to the findings. Results are not commented and it is very difficult to distinguish between the original contribution of the article and previous literature on the topic.

I recommend a check of the literature according to the mentioned points. The authors should revise not the written English, but clarify the meaning of technical terms used now and then, such as quantitative signature. They should also avoid the usage of ambiguous terms.

6. PLOS authors have the option to publish the peer review history of their article (what does this mean?). If published, this will include your full peer review and any attached files.

Reviewer #1: No

Reviewer #2: No

---

## [Decision Letter · Decision Letter 1]

23 Jun 2020

EEG state-trajectory instability and speed reveal global rules of intrinsic spatiotemporal neural dynamics

PONE-D-19-30275R1

Dear Dr. Suzuki,

We’re pleased to inform you that your manuscript has been judged scientifically suitable for publication and will be formally accepted for publication once it meets all outstanding technical requirements.

Kind regards,

Giovanni Petri, Ph.D.

Academic Editor

PLOS ONE

Additional Editor Comments (optional):

The reviewers were satisfied with the new version of the paper.

Reviewers' comments:

Reviewer's Responses to Questions

**Comments to the Author**

1. If the authors have adequately addressed your comments raised in a previous round of review and you feel that this manuscript is now acceptable for publication, you may indicate that here to bypass the “Comments to the Author” section, enter your conflict of interest statement in the “Confidential to Editor” section, and submit your "Accept" recommendation.

Reviewer #1: All comments have been addressed

Reviewer #3: All comments have been addressed

2. Is the manuscript technically sound, and do the data support the conclusions?

Reviewer #1: Yes

Reviewer #3: Yes

3. Has the statistical analysis been performed appropriately and rigorously? 

Reviewer #1: Yes

Reviewer #3: Yes

4. Have the authors made all data underlying the findings in their manuscript fully available?

Reviewer #1: Yes

Reviewer #3: Yes

5. Is the manuscript presented in an intelligible fashion and written in standard English?

Reviewer #1: Yes

Reviewer #3: Yes

6. Review Comments to the Author

Reviewer #1: (No Response)

Reviewer #3: the paper is interesting and well written, the reported methodology is appropriate, the conclusions appropriately supported

7. PLOS authors have the option to publish the peer review history of their article (what does this mean?). If published, this will include your full peer review and any attached files.

Reviewer #1: No

Reviewer #3: No

---

## [Editor Report · Acceptance letter]

17 Aug 2020

PONE-D-19-30275R1 

EEG state-trajectory instability and speed reveal global rules of intrinsic spatiotemporal neural dynamics 

Dear Dr. Suzuki:

I'm pleased to inform you that your manuscript has been deemed suitable for publication in PLOS ONE. Congratulations! Your manuscript is now with our production department. 

Kind regards, 

on behalf of

Dr. Giovanni Petri 

Academic Editor

PLOS ONE